# Graph Inverse Style Transfer for Counterfactual Explainability

**Bardh Prenkaj** [1 2 *]  **Efstratios Zaradoukas** [1 *]  **Gjergji Kasneci** [1]

## Abstract

Counterfactual explainability seeks to uncover model decisions by identifying minimal changes to the input that alter the predicted outcome. This task becomes particularly challenging for graph data due to preserving structural integrity and semantic meaning. Unlike prior approaches that rely on forward perturbation mechanisms, we introduce Graph Inverse Style Transfer (GIST), the first framework to re-imagine graph counterfactual generation as a backtracking process, leveraging spectral style transfer. By aligning the global structure with the original input spectrum and preserving local content faithfulness, GIST produces valid counterfactuals as interpolations between the input style and counterfactual content. Tested on 8 binary and multi-class graph classification benchmarks, GIST achieves a remarkable +7.6% improvement in the validity of produced counterfactuals and significant gains (+45.5%) in faithfully explaining the true class distribution. Additionally, GIST's backtracking mechanism effectively mitigates overshooting the underlying predictor's decision boundary, minimizing the spectral differences between the input and the counterfactuals. These results challenge traditional forward perturbation methods, offering a novel perspective that advances graph explainability.

## 1. Introduction

> *Explainability is no longer a luxury but a necessity.*
>
> (European Union, 2023)

High-stake domains such as healthcare (Amann et al., 2020), finance (Černevičienė & Kabašinskas, 2024), and digital forensics (Shamoo, 2025) have seen an abrupt interest in

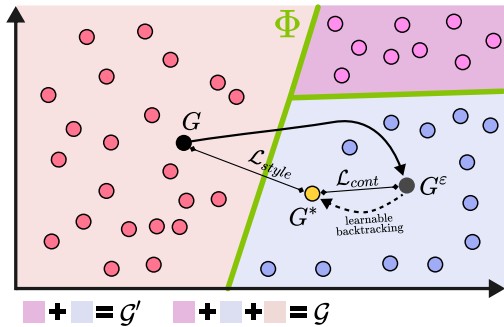

*Figure 1.* **The "gist" of GIST.** Starting from an initial graph $G \in \mathcal{G} \setminus \mathcal{G}'$, we overshoot to the other side of $\Phi$'s boundary on a randomly chosen graph $G^\varepsilon \in \mathcal{G}'$. We then learn a reverse process to backtrack from $G^\varepsilon$ to a never-seen-before graph $G^* \in \mathcal{G}'$ while imitating the global structure or style ($\mathcal{L}_{style}$) of $G$ and maintaining faithful local structure and preserve content ($\mathcal{L}_{cont}$) with $G^\varepsilon$. Notice how $\mathcal{L}_{style}$ pulls the generation towards $G$, satisfying the similarity condition in Equation (3). Meanwhile, $\mathcal{L}_{cont}$ pulls $G^*$ to $G^\varepsilon$ since we do not want to cross the decision boundary again and produce an invalid counterfactual.

equipping their users and service providers with explainable components, usually post-hoc, allowing them to make informed and reliable decisions (Guidotti et al., 2018). However, deep neural networks, commonly used for generating predictions, often suffer from a lack of interpretability, widely referred to as the *black-box* problem (Petch et al., 2021), hindering their wide adoption in these domains. Regulations such as the GDPR (European Union, 2016) and the EU AI Act (European Union, 2023) emphasize the demand for models that provide interpretable and actionable insights into their predictions. Alas, black-box models demonstrate superior performance and generalization capabilities when dealing with high-dimensional data to their inherently interpretable counterparts (Aragona et al., 2021; Diko et al., 2025; Flaborea et al., 2023a;b; Prenkaj et al., 2023).

Recently, graph neural networks (GNNs) (Scarselli et al., 2008) have achieved remarkable results in graph prediction tasks, such as community detection (Wu et al., 2022), link prediction (Wei et al., 2022), and session-based recommendations (Wu et al., 2019). Despite their remarkable performance, GNNs are black boxes, making them unsuitable for high-impact and high-risk scenarios. The literature

---
*Equal contribution [1] Technical University of Munich, Germany [2] Sapienza University of Rome, Italy. Correspondence to: Bardh Prenkaj <bardhprenkaj95@gmail.com>.

*Proceedings of the 42nd International Conference on Machine Learning*, Vancouver, Canada. PMLR 267, 2025. Copyright 2025 by the author(s).

has proposed several post-hoc explainability methods to understand *what is happening under the hood* of the prediction models. Counterfactual explanations (Wachter et al., 2017) have emerged as a key element in meeting regulatory requirements, as they shed light on model decisions by presenting alternative scenarios that would result in different outcomes. Furthermore, to support explanations for GNNs, a recent field in Graph Counterfactual Explainability (GCE) has emerged (Prado-Romero et al., 2023).

Existing solutions in GCE learn a forward[1] perturbation mechanism to produce a counterfactual w.r.t. an underlying decision model, hereafter called oracle $\Phi$. One obvious drawback of trying to forwardly cross $\Phi$'s decision boundary involves using $\Phi$ in the learning process (training) – e.g., see (Prado-Romero et al., 2024b) – which might not be feasible in scenarios where its predictions are limited by design.[2] A strictly forward learning approach can discard essential partial structure in the data by always starting from an uninformative initialization, rather than building upon already available signals. Without incremental corrective feedback, it often removes valid relationships or overshoots the decision boundary, resulting in suboptimal or oscillatory convergence. Furthermore, since this method lacks a mechanism to selectively preserve beneficial features during the transformation, it may converge to local minima that fail to retain important topological patterns. Consequently, the overall process can suffer from instabilities, inefficient exploration of parameter space, and diminished preservation of informative characteristics in the produced counterfactual.

To tackle these challenges, we propose GIST, short for **G**raph **I**nverse **S**tyle **T**ransfer, a transformative framework, inspired by the principles of style transfer in computer vision, that re-imagines counterfactual generation as a process of structural backtracking. By first overshooting $\Phi$'s decision boundary and then refining the graph towards a desirable configuration – i.e., the input graph's spectral properties (style) – GIST produces counterfactuals that preserve the original graph's global style and local node/edge content.

Specifically, we go beyond the related work by making the following contributions:

1. **First Graph Style Transfer Framework.** We propose GIST, a novel backtracking approach for graph counterfactual explainability. Unlike prior methods that forwardly cross the decision boundary, GIST leverages spectral style transfer to interpolate between global

structure alignment and local content preservation, ensuring semantically valid counterfactuals. We argue that this opens a new perspective in generating counterfactuals, as the overshooting factor (critical in forward approaches) is controllable via the learning objective.

2. **Theoretical Insights on Spectral Style Transfer.** We establish a rigorous theoretical foundation for GIST by analyzing the spectral properties of graph Laplacians. Specifically, we prove key results, including bounds on the spectral gap and Frobenius norm differences under convex combinations of Laplacians, ensuring stylistically (w.r.t. input) coherent counterfactuals.

3. **Scalable and Flexible Learning Framework.** GIST introduces a modular architecture that integrates transformer-based graph convolutional layers and differentiable edge sampling, enabling efficient backtracking. GIST supports a variety of GNN backbones, making it adaptable to diverse application domains while maintaining strong theoretical guarantees.

4. **Comprehensive Empirical Evaluation.** Extensive experiments on 8 benchmark datasets – spanning synthetic and real-world graphs with binary and multi-class classification tasks – emphasizes GIST as consistently outperforming SoTA. Specifically, GIST achieves considerably higher validity (+7.6% over the second-best) and improves fidelity by a large margin (+45.5%). Our results highlight GIST's ability to generate counterfactuals that are both more faithful and spectrally aligned (preserved semantics) with the input.

## 2. Preliminaries

**Graphs, their "style", and content.** Let $G = (X, A)$ be a graph consisting of node features $X \in \mathbb{R}^{n \times d}$ and an adjacency matrix $A \in \mathbb{R}^{n \times n}$ representing the connectivity among nodes with weights in the edges. We denote the graph dataset with $\mathcal{G} = \{G_1, \ldots, G_N\}$. The Laplacian matrix of a graph $G$ is defined as

$$L^{(G)} = D - A, \tag{1}$$

where $D$ is the degree matrix. The eigenvalues of $L^{(G)}$ are denoted as $\lambda_1(L^{(G)}) \leq \lambda_2(L^{(G)}) \leq \cdots \leq \lambda_n(L^{(G)})$. The normalized Laplacian is defined as

$$\tilde{L}^{(G)} = I - D^{-1/2} L^{(G)} D^{-1/2}, \tag{2}$$

with eigenvalues $\lambda_1(\tilde{L}^{(G)}) \leq \lambda_2(\tilde{L}^{(G)}) \leq \cdots \leq \lambda_n(\tilde{L}^{(G)})$. Notice how the content of the graph is encoded in the node feature vectors, and the style is encoded in the eigenvalues, summarizing global structural patterns. Laplacians capture global structural patterns, e.g., connectivity and symmetry, that are largely invariant to specific node identities. This follows a similar rationale to neural style transfer in images, where Gram matrices of feature activations are used

---

[1] Throughout this paper we use "forward" and "backward" in their literal sense, and not to refer to machine learning aspects. For the latter, we will refer to "forward learning pass" and "backward learning pass" to discern between the literal and ML meanings.

[2] Imagine deploying $\Phi$'s inference function as an API call. Calling these APIs at every training step would saturate the network bandwidth or, worse, get the caller IP blocked for ToS violations.

to model style since they encode correlation patterns among features rather than spatial arrangements. Additionally, using the Laplacian aligns with prior work in spectral graph theory, where the eigenvalues and eigenvectors of the Laplacian are shown to be robust descriptors of global structure, and have been used in graph matching (Yan et al., 2016) and generation (Dwivedi et al., 2023).

**Graph Counterfactuals.** Given a black-box (oracle) predictor $\Phi : \mathcal{G} \to Y$, according to (Prado-Romero et al., 2023), a counterfactual for $G$ is defined as

$$\underset{G' \in \mathcal{G}', \Phi(G) \neq \Phi(G')}{\arg \max} \mathcal{S}(G, G') \tag{3}$$

where $\mathcal{G}'$ is the set of all possible counterfactuals generated by perturbing $G$, and $\mathcal{S}(G, G')$ calculates the similarity between $G$ and $G'$. Notice how it is trivial to express Equation (3) in terms of graph distance instead of similarity function, which closely aligns with the counterfactual formalization in historically (Wachter et al., 2017), and more recently, (Leemann et al., 2024). Interestingly, Prenkaj et al. (2024) reformulate Equation (3) and take a probabilistic perspective to produce a counterfactual that is quite likely within the distribution of valid counterfactuals,

$$\underset{G' \in \mathcal{G}'}{\arg \max} P(G' \mid G, \Phi(G), \neg \Phi(G)), \tag{4}$$

where $\neg\Phi(G)$ indicates any other class from $\Phi(G)$.

## 3. Related Work

Our work directly relates to style transfer and graph counterfactual explainability, which we describe below. To the best of our knowledge, style transfer has never been defined or applied to graphs due to their inherent complex structures. However, for completeness purposes, we include the most interesting style transfer works in computer vision and natural language processing.

### 3.1. Style Transfer

**Computer Vision.** Gatys et al. (2016b) separate the content and style of images and combine them to generate new images. Gatys et al. (2016a) use a simple linear model to change the color of pictures by using a single image to represent the style. Zhu et al. (2017) propose CycleGAN to do image-image translation. It firstly learns a mapping $G : X \to Y$ using an adversarial loss, and then a reverse mapping $F : Y \to X$ with a cycle loss $F(G(X)) \approx X$ which performs unpaired image to image translation. Li et al. (2017) treat style transfer as a domain adaptation problem. They theoretically show that Gram metrics is equivalent to minimize the Maximum Mean Discrepancy (MMD) for images. While these approaches capture style by effectively

representing global texture, they lack flexibility in class-related structural properties. In contrast, GIST introduces a style representation that captures both local (input-specific) and global (class-related) structural patterns.

**Natural Language Processing.** Jhamtani et al. (2017) explore automatic methods to transform text from modern to Shakespearean English. Their model is based on seq2seq and enriched with pointer network (Vinyals et al., 2015). They use a modern-Shakespeare word dictionary to form candidate words for pointer network. However, paired-word dictionaries are scarce resources that do not exist in most style transfer tasks, requiring parallel corpora. Mueller et al. (2017) propose a variational autoencoder (VAE) to revise a new sequence to improve its associated outcome. Shen et al. (2017) explore style transfer for sentiment modification, decipherment of word substitution ciphers, and recovery of word order. They use a VAE as the base model and an adversarial network to align different styles. Braud & Søgaard (2017) explore many types of features for style prediction, ranging from n-grams to discourse, and found that simple models perform well. Ficler & Goldberg (2017) control linguistic style of generated text using conditioned recurrent neural networks (CRNNs). Fu et al. (2018) propose two models for text style transfer without parallel data: i.e., a multidecoder sequence-to-sequence model and a style embedding model. In line with (Fu et al., 2018), GIST finds counterfactuals without relying on parallel data (i.e., graphs whose style to imitate). Rather, we use the input to guide the overshooting mechanism and then act as a style which pulls the generation process towards it to preserve semantics.

### 3.2. Graph Counterfactual Explainability (GCE)

Prado-Romero et al. (2023) provide a detailed taxonomy of GCE methods composed of search-, heuristic- and learning-based approaches. Although a new category of global (model-level) counterfactual explanations is emerging (Huang et al., 2023; Kosan et al., 2024), our focus remains on instance-level and learning-based explainers. Although RCExplainer (Bajaj et al., 2021) is a mixture of heuristics and learned approaches – as per the taxonomy in (Prado-Romero et al., 2023) – we include it here to acknowledge its value with its multiple learned linear decision boundaries and then the search over these boundaries to find robust explanations.[3]

---

[3]Unfortunately, the official implementation could not be executed, as it depends on proprietary Huawei Python packages that are not publicly available. Despite extensive debugging and efforts to adapt the code to the GRETEL framework (Prado-Romero et al., 2024a) – in order to ensure consistency in the evaluation pipeline – we were unable to reproduce the original results. Additionally, we attempted to use an unofficial implementation available at `https://github.com/idea-iitd/gnn-x-bench/blob/main/source/rcexplainer.py`. However, this version lacks support for several benchmark datasets, including BBBP,

Learning-based strategies include perturbation matrices (Tan et al., 2022), reinforcement learning (Numeroso & Bacciu, 2021; Wellawatte et al., 2022), and generative approaches (Ma et al., 2022; Prado-Romero et al., 2024b). MEG (Numeroso & Bacciu, 2021) is a reinforcement learning approach that generates counterfactuals for input molecules. Its reward function integrates task-dependent regularization, influencing the policy to select actions that lead to valid molecules (Zhou et al., 2019). CF-GNNExp. (Lucic et al., 2022) learns a binary perturbation matrix to sparsify the adjacency matrix of the original graph $G$, guided by a sparse neural network (Srinivas et al., 2017). $CF^2$ (Tan et al., 2022) balances factual and counterfactual reasoning via multi-objective optimization. Counterfactuals are generated by removing the factual subgraph from the input and focusing on simplicity. CLEAR (Ma et al., 2022) employs a VAE to generate counterfactuals as complete graphs with stochastic edge weights, conditioned on the input graph and a desired class. During decoding, a graph matching step is required to address vertex reordering, an NP-hard problem (Livi & Rizzi, 2013). RSGG-CE (Prado-Romero et al., 2024b) relies on a modified learning approach of GANs and a partial-order sampling strategy on the learned edge distribution to generate robust graph counterfactual candidates. It exploits the generator to learn a graph representation, enabling stochastic estimations of the graph's topology, thus allowing the generation of counterfactuals in zero-shot. Unrelated to graphs, Nemirovsky et al. (2022) use GANs to generate counterfactuals for user-defined classes, adapted in (Prado-Romero et al., 2023) into G-CounteRGAN, treating adjacency matrices as black-and-white images with 2d convolutions.

The SoTA methods follow a forward perturbation paradigm (i.e., all learn how to cross $\Phi$'s decision boundary). To the best of our knowledge, GIST is the first that learns a backtracking mechanism (i.e., overshoot $\Phi$'s boundary first and then go backwards). By combining graph style transfer and counterfactual content preservation, GIST produces explanations that are spectrally and semantically aligned with the input, instead of merely similarity-wise – see Equation (3).

## 4. Method

We propose **GIST**[4] (short for **G**raph **I**nverse **S**tyle **T**ransfer), the first backtracking (inverse) mechanism towards the decision boundary of the oracle $\Phi$ instead of as-per-usual forwardly crossing it – see (Numeroso & Bacciu, 2021; Ma et al., 2022; Prado-Romero et al., 2024b) among others. In other words, given a graph $G$, our goal is to shoot over $\Phi$'s decision boundary and then move towards it in the opposite

direction without crossing it again. Figure 1 illustrates the idea behind our explainer. More formally, Definition 4.1 illustrates the conditions that counterfactual graphs found via style-transferring should meet.

**Definition 4.1.** Given a graph $G = (X, A)$, s.t. $n = |X|$, the goal is to generate $G^* = (X^*, A^*)$ by passing through an intermediary known graph $G^\varepsilon = (X^\varepsilon, A^\varepsilon) \in \mathcal{G}^*$ with $\Phi(G) \neq \Phi(G^\varepsilon)$, such that the following conditions are met.

(1) *Style transfer* – the global structural properties of $G^*$ align with $G$, i.e.,

$$\sum_{i=1}^n \left|\lambda_i(\tilde{L}^{(G)}) - \lambda_i(\tilde{L}^{(G^*)})\right| \leq \sum_{i=1}^n \left|\lambda_i(\tilde{L}^{(G)}) - \lambda_i(\tilde{L}^{(G^\varepsilon)})\right|, \quad (5)$$

(2) *Content preservation* – the local structure of $G^*$ resembles $G^\varepsilon$, i.e.,

$$\min_{G^* \in \mathcal{G}'} \quad \underbrace{\left||X^* - X^\varepsilon|\right|_1}_{\text{reconstruct node feature}} + \underbrace{\text{BCE}(A^*, A^\varepsilon)}_{\substack{\text{reconstruct existing} \\ \text{and non-existing edges}}} . \quad (6)$$

Appendix B shows a simple overshooting algorithm used in this paper.

### 4.1. Theoretical Implications

**Definition 4.2.** Given $\tilde{L}^{(G)}$ and $\tilde{L}^{(G^\varepsilon)}$, two real, symmetric and commuting matrices, the normalized Laplacian of $G^*$ is defined as the convex combination in Equation (7).

$$\tilde{L}^{(G^*)} = \alpha \cdot \tilde{L}^{(G^\varepsilon)} + (1 - \alpha) \cdot \tilde{L}^{(G)}, \quad (7)$$

where $\alpha \in [0, 1]$ is the interpolation factor that controls the trade-off between content and style.

**Lemma 4.3.** *Given $\tilde{L}^{(G)}$ and $\tilde{L}^{(G^\varepsilon)}$, two real, symmetric and commuting matrices, the eigenvalues of $G^*$ are defined as the convex combination*

$$\lambda_i(\tilde{L}^{(G^*)}) = \alpha \cdot \lambda_i(\tilde{L}^{(G^\varepsilon)}) + (1 - \alpha) \cdot \lambda_i(\tilde{L}^{(G)}). \quad (8)$$

Lemma 4.3 illustrates that the convexity of the Laplacian operator ensures that eigenvalues interpolate linearly. In other words, we show that the structure of the produced counterfactual $G^*$ is an interpolation (balanced by $\alpha$) between the structural style (i.e., from the input $G$) and the class-related spectrum (i.e., from $G^\varepsilon$).

Showing that the spectrum of $G^*$ is a linear interpolation of those of $G$ and $G^\varepsilon$ implies $G^*$ exhibits similar connectivity patterns to those of $G$ and $G^\varepsilon$. Hence, if $G$ and $G^\varepsilon$ are connected graphs (i.e., absence of isolated nodes), then also $G^\varepsilon$ is connected (see Theorem 4.4).

**Theorem 4.4.** *If $G$ and $G^\varepsilon$ are connected graphs, then $G^*$ whose normalized Laplacian is defined as in Equation (7) is connected for any $\alpha \in [0, 1]$.*

---

BZR, ENZYMES, MSRC21, and COLORS-3, hindering us to directly compare against other SoTA methods and GIST.

[4]Code: https://github.com/bardhprenkaj/gist

**Theorem 4.5.** *Let $\Delta(G) = \lambda_2(L^{(G)}) - \lambda_1(L^{(G)})$ denote the spectral gap of G, where $\lambda_2(L^{(G)})$ and $\lambda_1(L^{(G)})$ are the second- and first-lowest eigenvalues of G. Then:*

$$\min(\Delta(G), \Delta(G^\varepsilon)) \leq \Delta(G^*) \leq \max(\Delta(G), \Delta(G^\varepsilon)). \quad (9)$$

**Corollary 4.6.** *The Frobenius norm difference between $G^\varepsilon$ and $G^*$ is*

$$\left\|L^{(G^\varepsilon)} - L^{(G^*)}\right\|_F = (1-\alpha)\left\|L^{(G^\varepsilon)} - L^{(G)}\right\|_F. \quad (10)$$

Corollary 4.6 illustrates the bounded style similarity, i.e., the counterfactual $G^*$ cannot be arbitrarily dissimilar from the input, but is bounded to be similar to the difference of spectra between $G$ and $G^\varepsilon$ controllable through $\alpha$. In simpler words, if we imagine $G$, $G^*$, and $G^\varepsilon$ as points on a 1D-plane representing their spectra, then $G^*$ must be in-between $G$ and $G^\varepsilon$. Appendix A contains omitted proofs.

## 4.2. Learning to Backtrack

Here, we describe how our proposed GIST model (see Algorithm 1 for the forward pass of the network, and Figure 2 for GIST's architecture) facilitates a *reverse transformation* from a perturbed or intermediate graph $G^\varepsilon$ to a refined structure $G^*$ that better aligns with the target topology $G$. This process, which we call *learning to backtrack*, addresses the challenge of undoing distortions or noise, in our scenario $G^\varepsilon$, introduced into the original graph $G$ – a critical step in tasks such as graph denoising (Zhou et al., 2024), and domain adaptation (Cai et al., 2024).

### 4.2.1. ARCHITECTURE OVERVIEW

Our architecture relies on two key ideas: i.e., (1) transformer-based node embeddings, and (2) edge probability estimations to find $G^*$ starting from $G^\varepsilon$.

**Transformer-based Node Embeddings.** We employ transformer convolution layers (Shi et al., 2021) to learn node representations from the potentially noisy edges of $G^\varepsilon$. These layers are interleaved with ReLU activation functions which produce intermediate node embeddings for each vertex. These final node embeddings reflect the structural roles of nodes *despite* the "edge noise" present in $G^\varepsilon$.

**Edge Probability Prediction.** Using the learned node embeddings, we predict edge probabilities via a small multi-layer perceptron (MLP). Specifically, we concatenate the representations of any two nodes $(i, j)$ and feed them to the MLP to estimate the likelihood of an edge $(i, j)$ existing. This design effectively *backtracks* from the corrupted edges by selectively reintroducing or discarding edges based on how consistent they are with the learned embeddings.

*Differentiable Sampling and Backpropagation:* A key obstacle in learning graph structure end-to-end is the *discrete* nature of edge decisions. To circumvent this, we adopt the Gumbel-Softmax relaxation (Jang et al., 2017), enabling *continuous* approximations of binary sampling. Concretely, for an edge with predicted probability $p_{i,j}$, we introduce Gumbel noise $\Gamma$ and compute

$$\varrho_{i,j} = \sigma\big[(\log(p_{i,j}+\epsilon) - \log(1-p_{i,j}+\epsilon)+\Gamma)\,/\,T\big], \quad (11)$$

where $\sigma$ is the sigmoid function, $\epsilon$ is a small constant for numerical stability, and $T$ is the temperature parameter controlling the "hardness" of the sample. During training, these *soft* samples $\varrho_{i,j}$ remain *differentiable* w.r.t. the network parameters, thus allowing standard backpropagation to adjust node embeddings and edge probabilities. At inference time, one can threshold $\varrho_{i,j}$ (or sample via a Bernoulli draw) to yield a final, discrete adjacency matrix.

### 4.2.2. RECOVERY OF $G^*$

Once the edge probabilities are computed for each potential connection, GIST derives a new edge set[5] $E^*$ which is a Bernoulli sample over $\varrho_{i,j} \;\forall(i,j)$. In this way, we can compute the spectral loss – i.e., L1 distance between the eigenvalues – between the adjacency matrix $A^*$ induced on $E^*$ and the original adjacency matrix $A$ of $G$. Moreover, according to Definition 4.1, we also account for the content preservation between $G^*$ and $G^\varepsilon$. Therefore, given the graphs $G$ and $G^\varepsilon$, taken from the forward overshooting procedure, GIST optimizes the loss in Equation (12).

$$\mathcal{L} = \underset{\substack{G^*=(X^*,A^*),\\ A^*=g(|X^*|,E^*),\\ X^*,\varrho,E^*=f_\theta(G^\varepsilon)}}{\arg\min} \quad \alpha \underbrace{\left[\left\|X^* - X^\varepsilon\right\|_1 + \mathrm{BCE}(A^*, A^\varepsilon)\right]}_{\mathcal{L}_{cont}}$$

$$+ (1-\alpha)\underbrace{\left[\sum_{i=1}^{n}\left|\lambda_i(\tilde{L}^{(G)}) - \lambda_i(\tilde{L}^{(G^*)})\right|\right]}_{\mathcal{L}_{style}},$$
$$(12)$$

where $\mathrm{BCE}(A^*, A^\varepsilon)$ is defined in the second component of Equation (6), $\alpha \in [0,1]$ is the interpolation factor which pulls (pushes) towards (away from) the decision boundary of $\Phi$ w.r.t. $G$ ($G^\varepsilon$), $f_\theta$ is the our network architecture, and $g(n, E)$ produces an adjacency matrix in $\mathbb{R}^{n\times n}$ based on the edge set $E$.

## 5. Experiments

### 5.1. Experimental setup

We compare GIST to other SoTA learning-based explanation methods (i.e., CF-GNNExp. (Lucic et al., 2022), CF$^2$ (Tan et al., 2022), CLEAR (Ma et al., 2022), and RSGG-CE

---

[5]Here, we abuse the notation of a graph introduced in Section 2, and indicate a graph $G = (V, E)$ with its vertex set $V$ and edge set $E$.

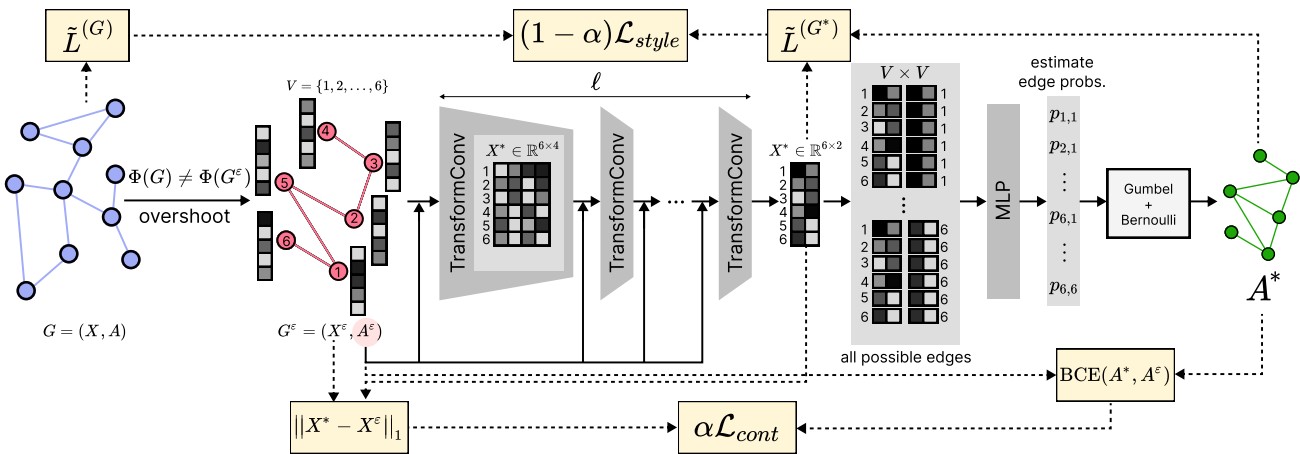

*Figure 2.* **Overview of GIST's forward learning pass and backtracking optimization**. Given a graph $G = (X, A)$, we use $\Phi$ to overshoot to $G^\varepsilon = (X^\varepsilon, A^\varepsilon)$ s.t. $\Phi(G) \neq \Phi(G^\varepsilon)$. We feed $G^\varepsilon$ through transformer convolution layers that output node embeddings $X^*$. For each embedding, we create edge pairs and feed them to an MLP which estimates probabilities $p_{i,j}$. We can sample these probabilities to compose a new adjacency matrix $A^*$. We maintain $G^\varepsilon$ faithfulness with $G^*$ by minimizing $\mathcal{L}_{cont}$, and spectral similarity with $G$ by minimizing $\mathcal{L}_{style}$. Overall GIST produces counterfactuals that strike a balance (linear interpolation) between $G$'s spectrum (style) and the local properties (content) of $G^\varepsilon$ via $\alpha\mathcal{L}_{cont} + (1 - \alpha)\mathcal{L}_{style}$.

*Table 1.* Average validity (the higher, the better) on the test set over 5-cross validations. Bold-faced digits show the best performing strategy; underline is the second-best. † depicts a binary-classification scenario; ‡ a multi-class scenario.

| | Real | | | | | | Synthetic | |
| --- | --- | --- | --- | --- | --- | --- | --- | --- |
| | AIDS † | BBBP † | BZR † | ENZYMES ‡ | MSRC21 † | PROTEINS † | BAShapes † | COLORS-3 ‡ |
| iRand | 0.013 | 0.151 | 0.332 | 0.134 | 0.035 | 0.018 | 0.000 | 0.392 |
| CF-GNNExp. | 0.936 | 0.931 | **0.810** | 0.910 | **0.965** | 0.378 | 0.516 | 0.736 |
| CF$^2$ | 0.019 | 0.208 | 0.185 | 0.437 | 0.018 | 0.039 | 0.000 | 0.676 |
| CLEAR | 0.037 | 0.267 | 0.176 | 0.370 | 0.933 | 0.563 | 0.908 | 0.217 |
| RSGG-CE | 0.128 | 0.404 | 0.732 | 0.447 | 0.912 | 0.237 | **1.000** | **0.884** |
| GIST | **0.969** | **0.956** | **0.810** | **0.970** | **0.965** | **0.791** | **1.000** | **0.884** |

*Table 2.* Average fidelity (the higher, the better) on the test set over 5-cross validations. Bold-faced digits show the best performing strategy; underline is the second-best. † depicts a binary-classification scenario; ‡ a multi-class scenario.

| | Real | | | | | | Synthetic | |
| --- | --- | --- | --- | --- | --- | --- | --- | --- |
| | AIDS † | BBBP † | BZR † | ENZYMES ‡ | MSRC21 † | PROTEINS † | BAShapes † | COLORS-3 ‡ |
| iRand | 0.013 | 0.177 | 0.146 | 0.004 | -0.035 | -0.009 | 0.000 | -0.007 |
| CF-GNNExp. | 0.924 | 0.784 | **0.741** | 0.077 | 0.825 | 0.202 | 0.484 | 0.091 |
| CF$^2$ | 0.015 | 0.178 | 0.176 | 0.017 | -0.018 | 0.018 | 0.000 | 0.065 |
| CLEAR | 0.050 | 0.164 | 0.127 | 0.040 | 0.855 | 0.051 | 0.060 | 0.033 |
| RSGG-CE | 0.124 | 0.286 | 0.683 | 0.050 | 0.807 | 0.133 | **0.968** | 0.147 |
| GIST | **0.957** | **0.809** | **0.741** | **0.203** | **0.860** | **0.425** | **0.968** | **0.202** |

(Prado-Romero et al., 2024b) in 8 benchmarking datasets (Appendix D) for graph classification with different evaluation metrics (Appendix E) relying on the GRETEL framework (Prado-Romero et al., 2023a; 2024a). We adapted RSGG-CE and CLEAR, originally available only for binary, to work in multi-class classifications scenarios. We rely on the iRand baseline (Prado-Romero et al., 2023b) to verify whether the SoTA actually learns to produce valid counterfactual explanations or are random perturbations sufficient. We use the default hyperparameters for the SoTA explainers,

**Algorithm 1** Forward learning pass of GIST

---

**Require:** $G^\varepsilon = (X^\varepsilon, A^\varepsilon)$ with vertex set $V$ and edge set
$E$, temperature $T$, noise $\Gamma$, model parameters $\theta$, number
of attention heads $h$, number of convolution layers $\ell$
**Ensure:** $X^*, \varrho_{i,j} \ \forall (i,j) \in V \times V, E^*$
1: $X^* \leftarrow X^\varepsilon$
2: **for** $i = 1 \ldots \ell$ **do**
3: $\quad X^* \leftarrow \text{ReLU}\big(\text{TransformerConv}_{i,\theta}(X^*, A^\varepsilon, h)\big)$
4: **end for**
5: $p_{i,j} \leftarrow \text{MLP}_\theta\big(X^*[i], X^*[j]\big) \ \forall (i,j) \in V \times V$
6: $\varrho_{i,j} \leftarrow \text{GumbelSoftmax}\big(p_{i,j}, T, \Gamma\big) \ \forall (i,j) \in V \times V$

7: $E^* \leftarrow \{(i,j) \mid \text{Bernoulli}\big(\varrho_{i,j}\big) = 1\} \ \forall \varrho_{i,j}$
8: **Return:** $X^*, \varrho, E^*$

---

and $p = .01$ and $t = 3$ for iRand (Appendix C). Unless
differently stated, we set $\alpha = 0.9$ for GIST. To generate
explanations, we first train the underlying oracles – three-
layered GCNs interleaved with ReLU activation functions
– and use the same weights for all explainers to ensure fair
performance comparisons. Table 5 – Table 12 illustrate the
accuracy of the oracles on the test sets of the datasets. We
use a 90:10 train-test split for all explainers and designate
10% of the training set as validation. We perform 5-fold
cross validations to assess the performances of the explain-
ers on one AMD EPYC 7002/3 64-Core CPU (for smaller
models) and one Nvidia TESLA V100 (for larger models)
totaling $\sim450h$ of execution time.

### 5.2. Results

**GIST has an average 7.6% gain over the second-best in
terms of validity, and 45.5% in fidelity.** Table 1 illustrates
the validity of all explainers on binary and multi-class graph
classification tasks. GIST outperforms SoTA explainers on
4/8 datasets (+13.3%), performs on par with CF-GNNExp.
on 2/8 and RSGG-CE on 2/8. Although validity measures
the effectiveness of the explainers crossing the decision
boundary, we also measure the fidelity of the produced
counterfactuals w.r.t. the oracle (see Table 2). Notice that
GIST, in COLORS-3, reports a better fidelity that RSSG-CE
(i.e., +37.4% improvement) although the latter has a similar
validity. Finally, GIST is on par with CF-GNNExp. on BZR
and MSRC21 in terms of validity. However, we report an
average gain of 2.12% on fidelity, hinting that we are more
faithful to the true class distribution (see Appendix E). We
show all measured performances of the explainers for all
datasets in Appendix G.

### 5.3. Ablation studies

**The interpolation factor $\alpha$ controls the trade-off between
the distances of $G$, $G^*$, and $G^\varepsilon$, and the overall validity.**

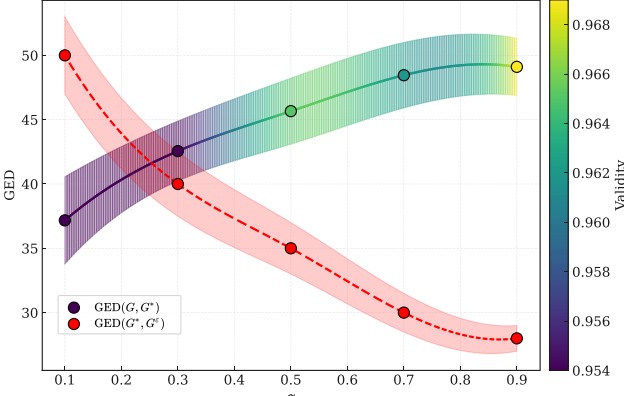

*Figure 3.* **As $\alpha \to 1$, the distance between $G$ and $G^*$ becomes
larger; the distance between $G^*$ and $G^\varepsilon$ decreases.** We show
the relationship between the interpolation factor $\alpha$, the GED, and
the validity of $G^*$ as guided by the oracle $\Phi$ on AIDS. Notice how
the validity of GIST increases when the distance between $G^*$ and
$G$ increases.

According to Figure 1 and the intuition of Corollary 4.6,
we expect that, regardless of the value of the interpolation
factor $\alpha$, the counterfactual $G^*$, if imagined as a point in a
1D-plane, must be in-between $G$ and $G^\varepsilon$. To support our
claim, we train GIST on 5-fold cross-validations on AIDS
with changing $\alpha \in \{0.1, 0.3, 0.5, 0.7, 0.9\}$ and measure the
GED and validity of the counterfactuals – see Figure 3.
When $\alpha \to 1$, the GED between $G$ and $G^*$ increases since,
according to Equation (12), GIST "prefers" to stay nearby
$G^\varepsilon$. As a consequence, we expect that the GED between
$G$ and $G^\varepsilon$ decreases when $\alpha \to 1$ (see the red curve). Fi-
nally, we show that if the GED between $G$ and $G^*$ is high,
then GIST must have overshot "far away" from $\Phi$'s bound-
ary to reach $G^\varepsilon$.[6] Hence, the chance of finding a correct
counterfactual is higher – i.e., GIST has a higher validity.

**GIST finds counterfactuals whose eigenvalues are closely
aligned to the optimal convex combination of Equa-
tion (8).** To demonstrate our theoretical findings, we in-
vestigate the eigenvalues of $G^*$ in accordance to the optimal
convex combination illustrated in Equation (8). Because
GIST is a learning approach which minimizes Equation (12),
it might happen that the spectrum of the produced counter-
factual $G^*$ is misaligned with the true combined spectra
of $G$ and $G^\varepsilon$. Figure 4 shows the spectra of 10 randomly
chosen $(G, G^\varepsilon)$ pairs from the AIDS dataset. Here, we il-
lustrate the eigenvalues of $G^*$ as produced by GIST, and
the true combined spectra with $\alpha = 0.9$. Notice how there
is a directly proportional relationship between the GED –
see Figure 3 – and the spectral combination of $G^*$. There-
fore, for $\alpha = 0.9$, we expect that the eigenvalues of $G^*$ are
more aligned with those of $G^\varepsilon$. Indeed, the figure confirms

---

[6]Conceptually, $\text{GED}(G, G^*) \leq \text{GED}(G, G^\varepsilon)$.

our intuition and shows how $G^*$ is closely aligned with the true spectra combination, reporting an average error of only $2.051 \times 10^{-3}$ for the reported samples (vs. $1.67 \times 10^{-3}$ for the whole dataset). Note that the zeros in the figure are due to the padding between $G^\varepsilon$ and $G$ to compute the spectral differences, which need the adjacency and degree matrices to be of same dimensions. To account for different graph sizes, in the future, we will explore the Wasserstein distance of the eigenvalues instead of L1.

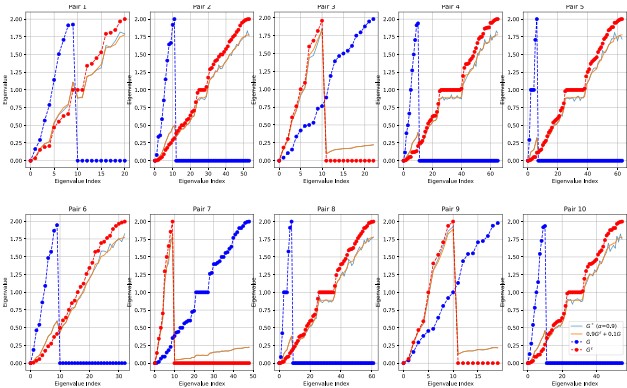

*Figure 4.* **GIST's backtracking mechanism finds $G^*$ with under $2.051 \times 10^{-3}$ error vs. the optimal counterfactual in terms of spectra convex combination in Equation (8).** We show the eigenvalues of 10 random $(G, G^\varepsilon)$ pairs on AIDS, and the finding of $G^*$ according to GIST with $\alpha = 0.9$. We also illustrate the optimal counterfactual produced w.r.t. the spectral convex combination $0.9L(G^\varepsilon) + 0.1L(G)$. Zeros in the figure are due to the padding between $G^\varepsilon$ and $G$ to compute the spectral differences.

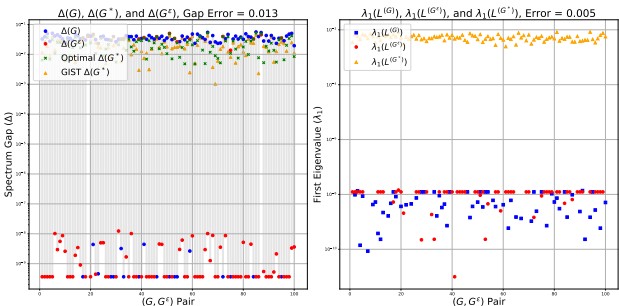

*Figure 5.* **GIST finds $G^*$ whose gap satisfies Theorem 4.5 with only $1.3 \times 10^{-2}$ difference against the optimal spectral gap.** (left) We show the spectral gaps of $G$, $G^\varepsilon$ and the found $G^*$ according to $\alpha = 0.9$ on 100 random samples of MSRC21. We also show the optimal counterfactual to assess the distance with $G^*$. (right) We show the first eigenvalue of $G$, $G^\varepsilon$, and $G^*$ to demonstrate the connectivity property in Theorem 4.4. We report an error of only $5 \times 10^{-3}$ from the optimal counterfactual. To show zeros in log-scale, we correct them with by adding the lowest gap/eigenvalue that is non-zero.

**GIST finds counterfactuals that respect the spectral gap of Theorem 4.5 and the connectivity property of The-**

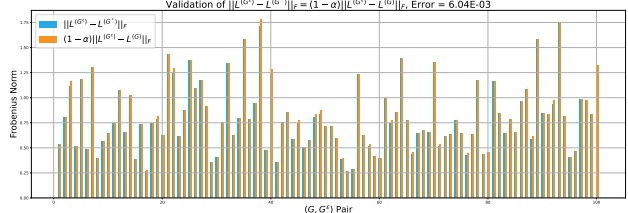

*Figure 6.* **GIST produces $G^*$ who are spectrally conformant to the style and local structure.** For $\alpha = 0.9$, we illustrate 100 randomly chosen $(G, G^\varepsilon)$ on PROTEINS, and measure the expected norm $(1 - \alpha)||L^{(G^\varepsilon)} - L^{(G)}||_F$. We also measure $||L^{(G^\varepsilon)} - L^{(G^*)}||_F$ on the predicted $G^*$ and report a negligible error of $6.04 \times 10^{-3}$.

**orem 4.4.** Figure 5 (left) shows the spectral gaps $\Delta(G)$, $\Delta(G^\varepsilon)$, and $\Delta(G^*)$ for 100 random samples of MSRC21 – see Appendix G.3.2 for more datasets. Notice that $\Delta(G^*)$ is always within the boundaries established in Theorem 4.5 and closely aligns with the optimal gap produced by the eigenvalues of the Laplacian from the convex combination $0.9L(G^\varepsilon) + 0.1L(G)$, reporting a gap error of only $1.3 \times 10^{-2}$. For completeness purposes, in Figure 5 (right) we illustrate the first eigenvalues of $G$, $G^\varepsilon$ and the produced $G^*$. Recall that a graph is connected if its first lowest eigenvalue is connected (i.e., $\lambda_1(L^{(G)}) = 0$). Notice how the majority of $G^\varepsilon$ graphs is connected – we corrected zero values to show in log-scale by adding the lowest eigenvalue that is non-zero – thus, depicting (mostly) a horizontal line. However, because $G$ is not always connected, then $G^*$ will also show $\lambda_1(L^{(G^*)}) > 0$. However, for fairness purposes, we notice that $\lambda_1(L^{(G)})$ and $\lambda_1(L^{(G^\varepsilon)})$ are only an epsilon away from zero (i..e, $\in [10^{-10}, 10^{-8}]$), practically resulting connected. Hence, as per Theorem 4.4, in the optimal scenario, we expect that the first eigenvalue of the counterfactual is 0. GIST generates $G^*$ whose connectedness is close to the optimal with $\lambda_1(L^{(G^*)}) \approx 5 \times 10^{-3}$. See Appendix G.3.2 for other datasets, and Figure 9 for a qualitative example of how two connected graphs $G$ and $G^\varepsilon$ produce a connected counterfactual $G^*$.

**GIST produces spectrally conformant counterfactuals.** We show that the produced counterfactuals are meaningful in terms of spectral similarity according to Corollary 4.6. Figure 6 illustrates the expected norm $(1 - \alpha)||L^{(G^\varepsilon)} - L^{(G)}||_F$ and the measured $||L^{(G^\varepsilon)} - L^{(G^*)}||_F$ from the predicted $G^*$. As shown in the theoretical part, we expect these two norm differences to be equal, showing that $G^*$ cannot be arbitrarily dissimilar to $G$ and $G^\varepsilon$. We argue that a spectral difference is a more principled measurement than the GED which is a mere edit distance metric of two graphs – i.e., it might happen that adding/removing edges moves the counterfactual away from the input, however it does not impact its spectrum. GIST reports a negligible

error of $6.04 \times 10^{-3}$ w.r.t. the scenario of producing the optimal counterfactual according to norm differences described above. It also reports lower differences between $G^\varepsilon$ and $G^*$, which implies that $G^*$ gives more importance to node and edge similarities rather than to $G$'s style properties. We suspect this arises due to the non-aggressive learning rate set for GIST which settles for a local minimum (see Appendix C). See Appendix G.3.3 for more experiments.

## 6. Limitations

**Trade-off Between Content and Style ($\alpha = 0.5$).** When equal weight is given to content and style, the model exhibits reduced performance. Specifically, it struggles to simultaneously preserve the original content structure while accurately matching the target style. This imbalance leads to significantly higher reconstruction errors compared to cases where one objective is prioritized. Detailed quantitative results supporting this observation are provided in Appendix G.4.

**No guarantees of class maintenance during backtracking.** After overshooting $G$ to $G^\varepsilon$ s.t. $\Phi(G) \neq \Phi(G^\varepsilon)$, the backtracking mechanism only guarantees that $\Phi(G) \neq \Phi(G^*)$. Nevertheless, while there are guarantees that $G^*$ has a structure which is an interpolation between the structural style ($G$) and the class-related spectrum ($G^\varepsilon$), it is not a given that $\Phi(G^*) = \Phi(G^\varepsilon)$ in a multiclass classification scenario. As mentioned in the next section, we leave this for further investigation in the future.

## 7. Conclusion

We introduced GIST, a novel method for graph counterfactual explainability by reformulating the task as a style transfer problem. We addressed key limitations of forward-based methods, such as compromising structural semantics, by leveraging spectral properties in a learnable backtracking mechanism. We integrated three core innovations: (1) style-transfer that preserves structural coherence while refining local node features and edge connections, (2) theoretical guarantees on spectral alignment and connectivity preservation, and (3) modular architecture compatible with diverse GNN backbones.

We highlighted GIST's superiority across eight benchmarks, achieving SoTA validity (+7.6%) and fidelity (+45.5%) on 8 benchmark datasets while maintaining computational efficiency. These results validate that backtracking, rather than forward traversal, enables precise control over counterfactuals, ensuring they remain structurally faithful to the input. We also showed that the interpolation factor guides the trade-off between distance and validity of the counterfactuals, whose spectral properties are within negligible empirical errors w.r.t. the optimal theoretical solution.

In the future, we will explore more efficient overshooting algorithms that help GIST choose a perturbed graph who is spectrally aligned to the input, thus improving convergence rates. Another avenue for investigation is to make theoretical guarantees that the counterfactuals maintain the class of the overshot graph. Lastly, we will explore global-level counterfactuality where the spectral commonalities of the class to explain can be used as guidance for the style transfer.

## Acknowledgements

Dr. Prenkaj was partially supported by the Lazio Region grant, FESR Lazio 2021 – 2027, project @HOME (# F89J23001050007, CUP B83C23006240002).

The authors would like to thank Roman Abramov for his acute observations during Dr. Prenkaj's seminar presentation that sparked the pursuit of the idea of style transfer in graph counterfactuality.

## Impact Statement

This paper presents work whose goal is to advance the field of Machine Learning. There are many potential societal consequences of our work, none which we feel must be specifically highlighted here.

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

# A. Omitted Proofs

## A.1. Lemma 4.3: Spectral Mixing Property

### A.1.1. COMMUTING CASE

*Proof.* The normalized Laplacians of $G$ and $G^\varepsilon$, $L^{(G)}$ and $L^{(G^\varepsilon)}$, commute and are simultaneously diagonalizable. Two matrices $L^{(G)}$ and $L^{(G^\varepsilon)}$ commute if $L^{(G)}L^{(G^\varepsilon)} = L^{(G^\varepsilon)}L^{(G)}$. If these matrices commute and are symmetric, there exists a single orthonormal matrix $U$ that diagonalizes both

$$L^{(G)} = U\Lambda U^\top, \ L^{(G^\varepsilon)} = U\Lambda^\varepsilon U^\top, \tag{13}$$

Following Equation (7), the styled Laplacian is therefore

$$L^{(G^*)} = \alpha \cdot U\Lambda^\varepsilon U^\top + (1-\alpha) \cdot U\Lambda U^\top. \tag{14}$$

Thus, we can approximate

$$L^{(G^*)} = U(\alpha\Lambda^\varepsilon + (1-\alpha)\Lambda)U^\top. \tag{15}$$

Since $\alpha\Lambda^\varepsilon + (1-\alpha)\Lambda$ is also a diagonal matrix, its diagonal entries are $\alpha\lambda_i^{(G^\varepsilon)} + (1-\alpha)\lambda_i^{(G)}$. Being diagonal in the same basis $U$, the eigenvalues of $L^{(G^*)} = \alpha L^{(G^\varepsilon)} + (1-\alpha)L^{(G)}$ are precisely

$$\left\{\alpha\lambda_i^{(G^\varepsilon)} + (1-\alpha)\lambda_i^{(G)}\right\}_{i=1}^n. \tag{16}$$

Thus, the eigenvalues interpolate linearly under convex combinations. The same reasoning holds for the normalized Laplacians $\tilde{L}^{(G)}$ and $\tilde{L}^{(G^\varepsilon)}$. $\square$

### A.1.2. NON-COMMUTING CASE

Although we assume that $L^{(G)}$ and $L^{(G^\varepsilon)}$ commute, we provide the reader with a complete overview when these matrices do not commute. Let their eigenvalues be sorted in non-decreasing order:

$$\lambda_1(L^{(G)}) \ \leq \ \lambda_2(L^{(G)}) \ \leq \ \cdots \ \leq \ \lambda_n(L^{(G)}), \tag{17}$$

$$\lambda_1(L^{(G^\varepsilon)}) \ \leq \ \lambda_2(L^{(G^\varepsilon)}) \ \leq \ \cdots \ \leq \ \lambda_n(L^{(G^\varepsilon)}). \tag{18}$$

Because $L^{(G)}$ and $L^{(G^\varepsilon)}$ do not commute, we cannot simply write $\lambda_i(L^{(G^*)}) = \alpha\lambda_i(L^{(G)}) + (1-\alpha)\lambda_i(L^{(G^\varepsilon)})$. Instead, we use Weyl's inequalities to bound $\lambda_k(L^{(G^*)})$.

**Courant-Fischer (Min-Max) Theorem.** Recall that for a real symmetric matrix $M$ with eigenvalues $\lambda_1(M) \leq \cdots \leq \lambda_n(M)$, the $k$-th eigenvalue $\lambda_k(M)$ can be characterized by:

$$\lambda_k(M) \ = \ \max_{\substack{U \subset \mathbb{R}^n \\ \dim(U)=k}} \ \min_{\substack{\mathbf{x} \in U \\ \mathbf{x} \neq \mathbf{0}}} \ \underbrace{\frac{\mathbf{x}^\top M \mathbf{x}}{\mathbf{x}^\top \mathbf{x}}}_{\text{Rayleigh quotient}}. \tag{19}$$

**Weyl's Inequalities: Statement**

**Theorem A.1** (Weyl's Inequalities)**.** *Let $A$ and $B$ be two real symmetric (or Hermitian) matrices of size $n \times n$. Denote their eigenvalues in non-decreasing order by*

$$\lambda_1(A) \leq \lambda_2(A) \leq \cdots \leq \lambda_n(A), \quad \lambda_1(B) \leq \lambda_2(B) \leq \cdots \leq \lambda_n(B). \tag{20}$$

*Let $C = A + B$. Then its eigenvalues $\{\lambda_i(C)\}_{i=1}^n$ also arranged in non-decreasing order satisfy:*

$$\lambda_k(A) + \lambda_1(B) \ \leq \ \lambda_k(A+B) \ \leq \ \lambda_k(A) + \lambda_n(B), \quad \text{for each } 1 \leq k \leq n. \tag{21}$$

*More precise interlacing bounds also exist, but this basic version often suffices to show that every eigenvalue of $A + B$ is contained within an interval determined by sums of the individual eigenvalues of $A$ and $B$.*

In our context, $A = \alpha L^{(G^\varepsilon)}$ and $B = (1-\alpha)L^{(G)}$. Since $A$ and $B$ are still real symmetric and $\alpha, (1-\alpha) \geq 0$, Weyl's theorem directly applies.

**Applying Weyl's Inequalities to $\alpha L^{(G^\varepsilon)} + (1 - \alpha)L^{(G)}$.** Putting $A = \alpha L^{(G^\varepsilon)}$ and $B = (1 - \alpha)L^{(G)}$ into Theorem Theorem A.1, we get for each $k = 1, \ldots, n$:

$$\lambda_k\big(\alpha L^{(G^\varepsilon)}\big) + \lambda_1\big((1 - \alpha)L^{(G)}\big) \;\leq\; \lambda_k\Big(\alpha L^{(G^\varepsilon)} + (1 - \alpha)L^{(G)}\Big) \;\leq\; \lambda_k\big(\alpha L^{(G^\varepsilon)}\big) + \lambda_n\big((1 - \alpha)L^{(G)}\big). \quad (22)$$

Using the fact that scaling a matrix by a nonnegative real factor $\alpha$ scales all its eigenvalues by $\alpha$, we have:

$$\lambda_k\big(\alpha L^{(G^\varepsilon)}\big) \;=\; \alpha\,\lambda_k\big(L^{(G^\varepsilon)}\big), \quad \lambda_j\big((1 - \alpha)L^{(G)}\big) \;=\; (1 - \alpha)\,\lambda_j\big(L^{(G)}\big). \quad (23)$$

Hence:

$$\alpha\,\lambda_k\big(L^{(G^\varepsilon)}\big) \;+\; (1 - \alpha)\,\lambda_1\big(L^{(G)}\big) \;\leq\; \lambda_k\Big(L^{(G^*)}\Big) \;\leq\; \alpha\,\lambda_k\big(L^{(G^\varepsilon)}\big) \;+\; (1 - \alpha)\,\lambda_n\big(L^{(G)}\big), \quad (24)$$

where $L^{(G^*)} = \alpha L^{(G^\varepsilon)} + (1 - \alpha)L^{(G)}$.

Therefore, while the $k$-th eigenvalue of $L^{(G^*)}$ does *not* equal a simple pointwise interpolation of $\lambda_k\big(L^{(G)}\big)$ and $\lambda_k\big(L^{(G^\varepsilon)}\big)$, it is constrained to lie within the interval:

$$\Big[\alpha\,\lambda_k\big(L^{(G^\varepsilon)}\big) + (1 - \alpha)\,\lambda_1\big(L^{(G)}\big), \;\; \alpha\,\lambda_k\big(L^{(G^\varepsilon)}\big) + (1 - \alpha)\,\lambda_n\big(L^{(G)}\big)\Big]. \quad (25)$$

*Remark* A.2 (Interpretation). This shows precisely why, in the non-commuting case, we only obtain *bounds* on the styled eigenvalues rather than a direct, index-by-index convex combination of the form $\alpha\,\lambda_k(L^{(G^\varepsilon)}) + (1 - \alpha)\,\lambda_k(L^{(G)})$. The lack of commutativity prevents simultaneous diagonalization. Therefore, the eigenvectors of $L^{(G^*)}$ differ from those of $L^{(G)}$ and $L^{(G^\varepsilon)}$, making a simple one-to-one mapping of eigenvalues impossible.

## A.2. Theorem 4.4: Connectedness Under Convex Combination of Laplacians

**Theorem A.3.** *Let $L^{(G)}$ and $L^{(H)}$ be the (combinatorial or normalized) Laplacians of two simple, undirected, **connected** graphs $G$ and $H$ on the same vertex set of size $n$. For any $\alpha \in [0, 1]$, define*

$$L^{(F)} := \alpha L^{(G)} + (1 - \alpha)L^{(H)}.$$

*Then $L^{(F)}$ corresponds to a connected graph.*

*Proof.* Assume for contradiction that the graph $F$ corresponding to $L^{(F)}$ is disconnected. Then there exists a nontrivial partition of the vertex set $V = S \cup (V \setminus S)$, with $S \neq \varnothing$ and $S \neq V$, such that no edge in $F$ connects a node in $S$ to a node in $V \setminus S$.

Let $A^{(G)}$ and $A^{(H)}$ denote the adjacency matrices of $G$ and $H$, respectively. Since $L^{(F)} = D^{(F)} - A^{(F)}$ and

$$A^{(F)} = \alpha A^{(G)} + (1 - \alpha)A^{(H)},$$

the absence of any edge crossing the cut in $F$ implies that for all $i \in S$, $j \in V \setminus S$,

$$A_{ij}^{(F)} = \alpha A_{ij}^{(G)} + (1 - \alpha)A_{ij}^{(H)} = 0.$$

Because $\alpha, 1 - \alpha \geq 0$, this implies that $A_{ij}^{(G)} = A_{ij}^{(H)} = 0$ for all such $(i, j)$. That is, neither $G$ nor $H$ has any edge crossing the cut between $S$ and $V \setminus S$. But this contradicts the assumption that both $G$ and $H$ are connected.

Therefore, such a cut cannot exist, and $F$ must be connected. Hence, $L^{(F)}$ is the Laplacian of a connected graph. $\qquad\square$

## A.3. Spectral Gap Bounds

**Theorem A.4** (Spectral Gap Bounds). *Let $G$ and $G^\varepsilon$ be two connected graphs on the same vertex set, and let $L^{(G)}$ and $L^{(G^\varepsilon)}$ be their (combinatorial or normalized) Laplacians. Define*

$$L^{(G^*)} := \alpha L^{(G^\varepsilon)} + (1 - \alpha)L^{(G)}, \quad \alpha \in [0, 1],$$

*and let the* spectral gap *be* $\Delta(G) := \lambda_2(L^{(G)})$, *with analogous definitions for $G^\varepsilon$ and $G^*$. Then,*

$$\min\big(\Delta(G), \Delta(G^\varepsilon)\big) \leq \Delta(G^*) \leq \max\big(\Delta(G), \Delta(G^\varepsilon)\big).$$

*Proof.* Since $G$ and $G^\varepsilon$ are connected, their Laplacians satisfy $\lambda_1 = 0 < \lambda_2$, and the same holds for $G^*$ by convexity and connectedness preservation (see Theorem A.3).

Let $A := \alpha L^{(G^\varepsilon)}$ and $B := (1-\alpha)L^{(G)}$, so $L^{(G^*)} = A + B$. By Weyl's inequality for symmetric matrices, we have:

$$\lambda_2(A + B) \in \big[\lambda_1(A) + \lambda_2(B),\ \lambda_2(A) + \lambda_1(B)\big].$$

Since $\lambda_1(L^{(G)}) = \lambda_1(L^{(G^\varepsilon)}) = 0$, this becomes:

$$\lambda_2(L^{(G^*)}) \in \big[\alpha\lambda_2(L^{(G^\varepsilon)}),\ (1-\alpha)\lambda_2(L^{(G)})\big].$$

Swapping $G$ and $G^\varepsilon$ gives the symmetric interval:

$$\lambda_2(L^{(G^*)}) \in \Big[\min\big(\alpha\lambda_2(L^{(G^\varepsilon)}), (1-\alpha)\lambda_2(L^{(G)})\big),\ \max\big(\alpha\lambda_2(L^{(G^\varepsilon)}), (1-\alpha)\lambda_2(L^{(G)})\big)\Big].$$

However, since $\alpha \in [0,1]$, the maximum of $\lambda_2(L^{(G^\varepsilon)})$ and $\lambda_2(L^{(G)})$ always bounds $\lambda_2(L^{(G^*)})$ from above, and the minimum from below. Thus,

$$\min\big(\Delta(G), \Delta(G^\varepsilon)\big) \leq \Delta(G^*) \leq \max\big(\Delta(G), \Delta(G^\varepsilon)\big).$$

$\square$

*Remark* A.5. For normalized Laplacians, the same bounds apply since they are also symmetric and positive semi-definite, and Weyl's inequality holds for all real symmetric matrices.

### A.4. Frobenius Norm Difference Corollary

**Corollary A.6.** *Let $L^{(G)}, L^{(G^\varepsilon)}, L^{(G^*)}$ be Laplacians (combinatorial or normalized) of graphs $G$, $G^\varepsilon$, and $G^*$ respectively, where*

$$L^{(G^*)} = \alpha L^{(G^\varepsilon)} + (1-\alpha) L^{(G)}, \quad \alpha \in [0,1].$$

*Then, the Frobenius norm difference between $G^\varepsilon$ and $G^*$ satisfies*

$$\big\|L^{(G^\varepsilon)} - L^{(G^*)}\big\|_F = (1-\alpha)\big\|L^{(G^\varepsilon)} - L^{(G)}\big\|_F.$$

*Proof.* By linearity, we have

$$L^{(G^\varepsilon)} - L^{(G^*)} = L^{(G^\varepsilon)} - \big(\alpha L^{(G^\varepsilon)} + (1-\alpha)L^{(G)}\big) = (1-\alpha)\big(L^{(G^\varepsilon)} - L^{(G)}\big).$$

Taking Frobenius norms and using absolute homogeneity gives the result:

$$\big\|L^{(G^\varepsilon)} - L^{(G^*)}\big\|_F = (1-\alpha)\big\|L^{(G^\varepsilon)} - L^{(G)}\big\|_F.$$

$\square$

## B. GIST's Overshooting Algorithm

Figure 2 illustrates the architecture of GIST and how it learns to backtrack (see Algorithm 1) into finding the counterfactuals $G^*$. Notice that GIST, before performing the backtracking mechanism, needs to overshoot the decision boundary of $\Phi$ given an input graph $G$. In this paper, given $G$, we rely on a simple overshooting algorithm which finds $G^\varepsilon$ as in Equation (26).

$$k^* = \min\left\{k \in [1,n] \mid \Phi(G) \neq \Phi(G_k)\right\} \quad \forall G_k \in \mathcal{U}(\{G \mid G \in \mathcal{G}),$$

$$G^\varepsilon := G_{k^*},$$

(26)

where $\mathcal{U}(*)$ depicts a shuffling operation from the set given in input, and $\mathcal{G}$ is the set of graphs in the dataset. An interesting overshooting algorithm, which would aid convergence and reduce the estimation errors shown in the main material, is that of choosing $G^{\varepsilon}$ given $G$ as in Equation (27).

$$k^* = \min \left\{ k \in [1, n] \mid \Phi(G) \neq \Phi(G_k) \wedge (1 - \alpha) \left\| L^{(G_k)} - L^{(G)} \right\|_F \right\} \quad \forall G_k \in \mathcal{U}(\{G \mid G \in \mathcal{G}\}),$$

$$G^{\varepsilon} := G_{k^*}.$$

(27)

Notice how we condition the selection of $G^{\varepsilon}$ on its Frobenius norm difference with $G$. This ensures that the chosen pair is structurally similar yet belonging to different classes. In this case, GIST would have more chances of converging to the optimal counterfactual $G^*$ than being agnostic to the choice of $G^{\varepsilon}$ as in Equation (26). We leave these investigations for future work.

## C. Hyperparameter Selection

In our experimental evaluation, we tested GIST against other learning-based explanation methods from the literature. The competitors include RSGG-CE (Prado-Romero et al., 2024b), CF$^2$ (Tan et al., 2022), CLEAR (Ma et al., 2022), and CF-GNNExp. (Lucic et al., 2022). For each explainer, we used the optimal hyperparameters proposed in their respective literature:

- For GIST we configured it to run the backtracking process for 50 epochs with a batch size of 16. We chose the number of attention heads to be equal to 2, the node embedding dimension to 16. We set $\alpha = 0.9$ to encourage higher validity, which is beneficial for a helpful counterfactual. We train GIST with Adam optimizer with learning rate $10^{-3}$ and a weight decay of $10^{-5}$.
- For CF$^2$ (Tan et al., 2022), we configured: 20 epochs, batch size ratio of 0.2, learning rate (lr) initialized at 0.02, and regularization parameters $\alpha = 0.7$, $\lambda = 20$, and $\gamma = 0.9$.
- CF-GNNExp (Lucic et al., 2022) utilized: $\alpha = 0.01$, $K = 5$, $\beta = 0.6$, and $\gamma = 0.2$.
- CLEAR (Ma et al., 2022) employed: 10 epochs, learning rate (lr) of 0.01, counterfactual loss regularization parameter ($\lambda_{\text{cfe}}$) set to 0.1, trade-off parameter $\alpha = 0.4$, and batch size 32.
- RSGG-CE (Prado-Romero et al., 2024b) was trained for 500 epochs with a GAN configuration: batch size 1 and *TopKPooling* discriminator.

Concerning the oracle implementation, we used the following hyperparameters: 50 epochs, batch size 32, and early stopping threshold $10^{-4}$. We trained the model using the RMS Propagation optimizer (learning rate lr $= 0.01$) with Cross Entropy loss. The architecture consisted of a Graph Convolutional Neural Network with 3 convolutional layers and 1 dense layer, convolutional booster 2, and linear decay factor 1.8.

## D. Dataset description

*Table 3.* Summary of selected datasets and their characteristics. † depicts a real-world dataset and ‡ a synthetic one.

| Dataset | Graphs | Classes | Avg. Nodes | Avg. Edges | Node Labels | Edge Labels | Node Attr. |
|---|---|---|---|---|---|---|---|
| AIDS † | 2000 | 2 | 15.69 | 16.20 | ✓ | ✓ | ✓(4) |
| BAShapes ‡ | 500 | 2 | 57.00 | 82.01 | – | – | ✓(8) |
| BBBP † | 2039 | 2 | 24.06 | 26.00 | ✓ | – | – |
| BZR † | 405 | 2 | 35.75 | 38.36 | ✓ | – | ✓(3) |
| COLORS-3 ‡ | 10500 | 11 | 61.31 | 91.03 | – | – | ✓(4) |
| ENZYMES † | 600 | 6 | 32.63 | 62.14 | ✓ | – | ✓(18) |
| MSRC21 † | 563 | 20 | 77.52 | 198.32 | ✓ | – | ✓(3) |
| PROTEINS † | 1113 | 2 | 39.06 | 72.82 | ✓ | – | ✓(1) |

We conducted our experiments on 8 popular real and synthetic datasets comprising of binary and multi-class graph classification tasks to compare our approach with SoTA GCE methods. Table 3 illustrates the characteristics of the datasets used in this paper.

- **AIDS** (Riesen & Bunke, 2008) consists of graphs representing molecular compounds. These graphs are derived from the AIDS Antiviral Screen Database of Active Compounds. This data set consists of two classes (active and inactive), which represent molecules with or without activity against HIV. The molecules are converted into graphs in a straightforward manner by representing atoms as nodes and the covalent bonds as edges. Nodes are labeled with the number of the corresponding chemical symbol and edges by the valence of the linkage. There are 2,000 elements in total (1,600 inactive elements and 400 active elements).

- **BAShapes** (Ying et al., 2019) is a synthetic dataset consisting of a base graph and motifs connected on the base. The base graph is a Barabasi-Albert (BA) graph and motifs can be either a house-shaped (class 0) or a grid-shape (class 1). Following the generation done in (Lucic et al., 2022), there are 8 nodes on the base graph with 5 edges connecting them. Each base graph has 7 motives connected to it.

- **BBBP** (Blood-Brain Barrier Penetration) (Martins et al., 2012) is a dataset widely used in drug discovery and neurological research to develop machine learning models that predict blood-brain barrier permeability. The blood-brain barrier is a protective membrane that shields the central nervous system by regulating the passage of solutes. Its presence is a critical consideration in drug development, whether for designing molecules that target the central nervous system or for identifying compounds that should be restricted from crossing the barrier. BBBP contains binary labels for 2,053 curated molecules, indicating whether a compound can penetrate the blood-brain barrier. Specifically, 1,570 molecules can penetrate the barrier, while 483 cannot.

- **BZR** (Sutherland et al., 2003) represents a set of 405 ligands for the benzodiazepine receptor. No differentiation of agonists, antagonists, and inverse agonists is made. In vitro binding affinities as measured by inhibition of [$^3$H] diazepam binding are expressed as $IC_{50}$ values, ranging from 0.34 nM to $> 70\mu$M (65 compounds have indeterminate values). The authors selected $pIC_{50} = 7.0$ as the threshold for activity by considering a histogram plot of compound counts vs $pIC_{50}$ and the resulting balance of active and inactive compounds.

- **COLORS-3** (Knyazev et al., 2019) is a synthetic graph benchmark designed to evaluate the interpretability and generalization capabilities of graph-based machine learning models, particularly those leveraging attention mechanisms. Each graph consists of nodes assigned one of three colors, with the task framed as a classification problem: models must predict the count of nodes belonging to a specific target color. For generalization purposes, the dataset includes training graphs of moderate size and test sets containing graphs that are significantly larger and more complex than those seen during training.

- **ENZYMES** (Borgwardt et al., 2005) consists of protein graph models. The dataset includes 600 enzymes from the BRENDA database (Schomburg et al., 2004), with 100 proteins sampled from each of the six Enzyme Commission (EC) top-level classes. The original objective that the dataset was created for is to accurately predict the enzyme class membership for these proteins.

- **MSRC21** (Neumann et al., 2016) is a state-of-the-art dataset in semantic image processing originally introduced in (Winn et al., 2005). Each image is represented by a conditional Markov random field graph, as illustrated in Figure 7. The nodes of each graph are derived by over-segmenting the images using the quick shift algorithm,[7] resulting in one graph among the superpixels of each image. Nodes are connected if the superpixels are adjacent, and each node can further be annotated with a semantic label. Imagining an image retrieval system, where users provide images with semantic information, it is realistic to assume that this information is only available for parts of the images, as it is easier for a human annotator to label a small number of image regions rather than the full image. As the images in the MSRC dataset are fully annotated, the authors derive semantic (ground-truth) node labels by taking the mode ground-truth label of all pixels in the corresponding superpixel. Semantic labels are, for example, building, grass, tree, cow, sky, sheep, boat, face, car, bicycle, and a label void to handle objects that do not fall into one of these classes.

- **PROTEINS** is a dataset created by Borgwardt et al. (2005), originally proposed in (Dobson & Doig, 2003), comprising 1,178 proteins (59% enzymes, 41% non-enzymes). Proteins were selected to ensure low structural similarity. From this set, 1,128 proteins (retaining the original class balance) were retained based on the availability of secondary structure data in the Protein Data Bank (PDB). In their original study, Dobson and Doig represented proteins using feature vectors. These vectors encoded various attributes, including the fraction of each amino acid (AA) type among all residues, the fraction of the protein's surface area occupied by each AA, the presence of ligands, the size of the largest

---

[7]https://www.vlfeat.org/overview/quickshift.html.

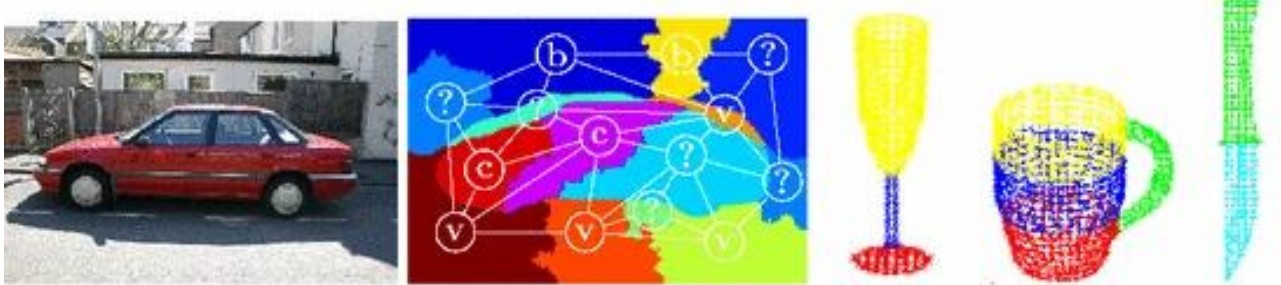

*Figure 7.* Taken from (Neumann et al., 2016). The left-most RGB image is represented by a graph of superpixels (middle) with semantic labels b *building*, c *car*, v *void*, and ? *unlabeled*. (right) Point clouds of household objects represented by labeled 4-nearest-neighbor graphs with part labels top (yellow), middle (blue), bottom (red), usable-area (cyan), and handle (green). Edge colors are derived from the adjacent nodes.

surface pocket, and the number of disulfide bonds. But, afterwards Borgwardt modeled proteins as graphs. This dataset is notably challenging due to its strict non-redundancy constraint, emphasizing generalizable functional prediction over sequence or structural homology.

## E. Evaluation Metrics

We adopt the evaluation framework proposed by (Prado-Romero et al., 2023), employing a diverse set of metrics for a comprehensive and fair evaluation. Our evaluation criteria include *Runtime*, *Oracle Calls* (Abrate & Bonchi, 2021), *Validity* (Guidotti, 2022; Prado-Romero et al., 2023a), *Sparsity* (Prado-Romero et al., 2023a; Yuan et al., 2022), *Fidelity* (Yuan et al., 2022), Oracle Accuracy, and *Graph Edit Distance* (GED) (Prado-Romero et al., 2023). Recall that $\Phi : \mathcal{G} \to Y$ is an oracle.

**Runtime** measures the time the explainer takes to generate a counterfactual example. This metric offers an efficient means of evaluating the explainer's performance, encompassing the execution time of the oracle. To ensure fairness, runtime evaluations must be conducted in isolation on the same hardware and software platform.

**Oracle Calls** (Abrate & Bonchi, 2021) quantifies the number of times the explainer queries the oracle to produce a counterfactual. This metric, akin to runtime, assesses the computational complexity of the explainer, especially in distributed systems. It avoids considering latency and throughput, which are external factors in the measurement.

**Oracle Accuracy** evaluates the reliability of the oracle in predicting outcomes. The accuracy of the oracle significantly impacts the quality of explanations, as the explainer aims to elucidate the model's behaviour. Mathematically, for a given input $G$ and true label $y$, accuracy is defined as $\chi(G) = \mathbb{I}[\Phi(G) = y]$.

**Validity** (Guidotti, 2022; Prado-Romero et al., 2023a) assesses whether the explainer produces a valid counterfactual explanation, indicating a different classification from the original instance. Formally, for the original instance $G$, the counterfactual $G'$, and oracle $\Phi$, validity is an indicator function $\Omega(G, G') = \mathbb{I}[\Phi(G) \neq \Phi(G')]$.

**Sparsity** (Yuan et al., 2022) gauges the similarity between the input instance and its counterfactual concerning input attributes. If $\mathcal{S}(G, G') \in \mathbb{R}_0^1$ is the similarity between $G$ and $G'$, we adapt the sparsity definition to $\frac{1-\mathcal{S}(G,G')}{|G|}$ for graphs.

**Fidelity** (Yuan et al., 2022) measures the faithfulness of explanations to the oracle, considering validity. Given the input $G$, true label $y$, and counterfactual $G'$, fidelity is defined as $\Psi(G, G') = \chi(G) - \mathbb{I}[\Phi(G') = y]$. Fidelity values can be **1** for the correct explainer and oracle, or **0** and **−1** indicating issues with either the explainer or the oracle.

**Graph Edit Distance (GED)** quantifies the structural distance between the original graph $G$ and its counterfactual $G'$. The distance is evaluated based on a set of actions $\{p_1, p_2, \ldots, p_n\} \in \mathcal{P}(G, G')$, representing a path to transform $G$ into $G'$. Each action $p_i$ is associated with a $\omega(p_i)$ cost. GED is computed as

$$\min_{\{p_1,\ldots,p_n\}\in\mathcal{P}(G,G')} \sum_{i=1}^{n} \omega(p_i)$$

Preference is given to counterfactuals closer to the original instance $G$, as they provide shorter action paths on $G$ to change

the oracle's output. GED offers a global measure and can be complemented by a relative metric like sparsity to assess the explainer's performance across instances.

## F. Time Complexity Analysis for GIST

In this section, we elaborate on the computational complexity of the calculations presented in Section 2. We identify four key components that require explicit complexity analysis: the Laplacian matrix $L(G)$, its normalized form $\tilde{L}(G)$, and their respective computations. Notice that we are not analyzing the forward learning pass of the backward mechanism illustrated in Figure 2. The time complexity of the forward learning pass is dominated by the calculation of the following matrices.

Given an undirected weighted graph $G$ with $n$ nodes and $m$ edges, the time complexities of the essential operations are as follows:

**Degree Matrix Computation $D$.** The degree matrix $D$ is a diagonal matrix where each diagonal element $D_{ii}$ represents the degree of node $i$. The degree is computed as the sum of the weights of the edges connected to node $i$, which corresponds to the row or column sum in the adjacency matrix $A$.

For dense graphs, where most nodes are highly connected, the adjacency matrix $A$ contains a significant number of non-zero entries. Computing the degree matrix by summing all entries in $A$ requires $O(n^2)$ time. However, for sparse graphs where $m \ll n^2$, only the non-zero entries (corresponding to edges) are considered, reducing the time complexity to $O(m)$. Therefore, the complexity of computing the degree matrix is:

$$O(n^2) \quad \text{for dense graphs}, \quad O(m) \quad \text{for sparse graphs}.$$

**Laplacian Matrix Formation $L(G) = D - A$.** The Laplacian matrix $L(G)$ is computed by subtracting the adjacency matrix $A$ from the degree matrix $D$. Both matrices have dimensions $n \times n$. In dense graphs, element-wise subtraction takes $O(n^2)$ time, since every entry must be processed. However, in sparse graphs, the adjacency matrix $A$ contains only $m$ non-zero entries, and $D$ is diagonal. Consequently, the subtraction operation can be performed in $O(n + m)$ time. The resulting complexity is:

$$O(n^2) \quad \text{for dense graphs}, \quad O(n + m) \quad \text{for sparse graphs}.$$

**Eigenvalue Decomposition.** Eigenvalue decomposition is crucial for analyzing the spectral properties of the Laplacian matrix. The standard approach, such as QR decomposition, has a time complexity of $O(n^3)$ for dense graphs.

For large sparse graphs, computing all eigenvalues directly is impractical. Instead, iterative methods like the Lanczos algorithm are employed, which approximate the largest (or smallest) eigenvalues efficiently. The Lanczos algorithm operates in $O(n + m)$ time per iteration without reorthogonalization, making it a practical alternative for sparse graphs. Thus, the time complexity of eigenvalue decomposition is:

$$O(n^3) \quad \text{for dense graphs}, \quad O(k(n + m)) \quad \text{for sparse graphs, where k denotes the iterations/eigenvalues}.$$

**Normalized Laplacian Computation $\tilde{L}(G) = D^{-1/2}L(G)D^{-1/2}$.** The normalized Laplacian $\tilde{L}(G)$ is obtained by applying the diagonal matrix $D^{-1/2}$ to both sides of the Laplacian matrix. Inverting the diagonal elements of $D$ to compute $D^{-1/2}$ takes $O(n)$ time, as it involves element-wise inversion.

For the matrix multiplication involved in forming $\tilde{L}(G)$, we need to multiply first $D^{-1/2}$ with $L(G)$, which has a complexity of $O(n^2)$, because $D^{-1/2}$ is a diagonal matrix.[8] Then, there is another multiplication step between $D^{-1/2}L(G)$ and $D^{-1/2}$, which also has a complexity of $O(n^2)$. Thus, the overall time complexity is $O(n^2)$ in dense graphs. However, for sparse graphs, the multiplication can be optimized to $O(m)$, since most elements in $L(G)$ are zero, allowing efficient computation. Thus, the time complexity is:

$$O(n^2) \quad \text{for dense graphs}, \quad O(m) \quad \text{for sparse graphs}.$$

This analysis demonstrates that our method does not introduce significant computational bottlenecks. As shown in Table 4, utilizing sparse representations significantly improves computational efficiency. For large sparse graphs, iterative methods for eigenvalue decomposition further reduce complexity to $O(n + m)$ per iteration, making the calculations feasible.

---

[8]Notice that if $D$ were not a diagonal matrix, the matrix multiplication would have a complexity of $O(n^3)$.

*Table 4.* Overall complexity considerations for dense and sparse graphs.

| Computation | Dense Graphs | Sparse Graphs |
|---|---|---|
| $D$ | $O(n^2)$ | $O(m)$ |
| $L(G) = D - A$ | $O(n^2)$ | $O(n + m)$ |
| Eigenvalue Decomposition | $O(n^3)$ | $O(k(n + m))$ (for k eigenvalues) |
| $\tilde{L}(G) = D^{-1/2}L(G)D^{-1/2}$ | $O(n^2)$ | $O(m)$ |

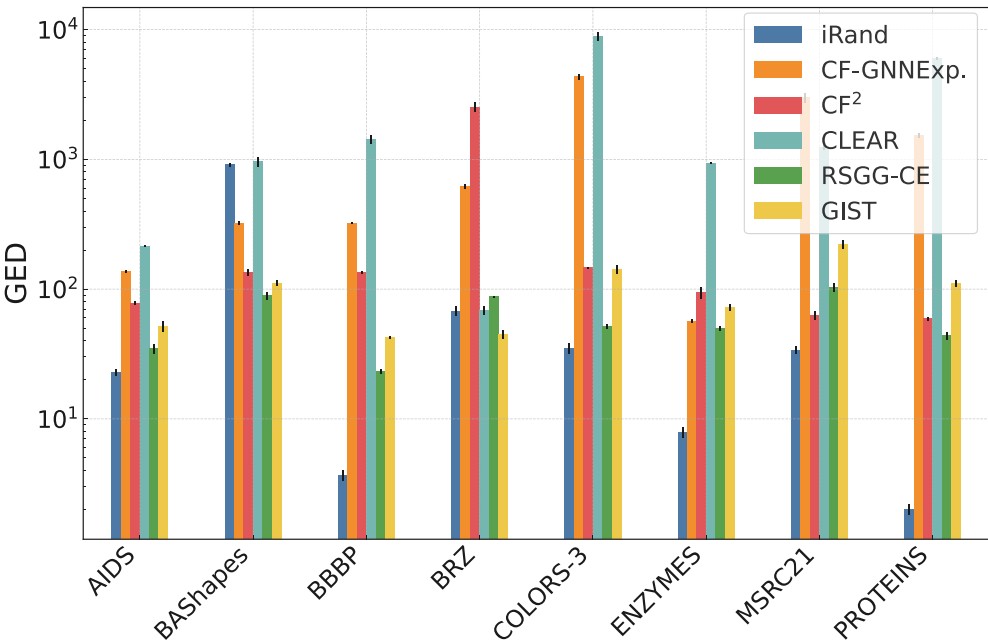

*Figure 8.* **GIST is the most effective and reliable explainer (in terms of validity), reporting low input structural changes.** Comparison of GED across different explainers for generating counterfactuals on various datasets. GIST consistently achieves the best balance between validity and structural proximity, with significantly lower GED than other methods w.r.t. the baseline iRand.

## G. More Experiments

### G.1. GIST vs. SoTA in terms of GED

**GIST maintains reasonable Graph Edit Distance (GED) over the low-validity iRand baseline with 8.9% increase against 2.1% reported by RSGG-CE.** Figure 8 illustrates the structural proximity in counterfactual generation, as measured by the GED. We invite the reader to assess the trade-off between validity and fidelity, shown in Table 1 and Table 2, and the GED. While RSGG-CE demonstrates lower GED in many cases (indicating smaller structural changes), it often sacrifices validity, which undermines its practical effectiveness in generating meaningful counterfactuals. Contrarily, GIST prioritizes generating valid counterfactuals, even at the cost of slightly higher GED. This approach aligns with the goal of counterfactual generation – providing actionable, interpretable, and valid examples – since low GED alone is insufficient if the counterfactuals are not valid. Here, GIST strikes a crucial balance by producing counterfactuals that remain meaningful and interpretable while requiring reasonable structural changes. This makes it a more dependable choice for real-world applications where validity is essential. For instance, in datasets like BZR and MSCR21 – where GIST performs comparably to CF-GNNExp in terms of validity – GIST achieves a GED that is approximately two orders of magnitude lower than CF-GNNExp. Overall, GIST reports an increase of 8.9% over iRand on all datasets; RSGG-CE 2.1%, CLEAR 776.2%, CF$^2$ 180.8%, and CF-GNNExp. 366.7%. This highlights GIST's effectiveness as a superior explainer that optimizes both validity and structural proximity.

*Table 5.* Average test-set performances on AIDS for 5-cross validations. The accuracy for the used GCN oracle in the test set is 99.4%.

|  | GED ↓ | Oracle Calls ↓ | Validity ↑ | Sparsity ↓ | Fidelity ↑ |
|---|---|---|---|---|---|
| iRand | $\mathbf{0.23 \times 10^1}$ | $\underline{1.10 \times 10^1}$ | 0.013 | **0.001** | 0.013 |
| CF-GNNExp. | $1.36 \times 10^2$ | $\mathbf{1.00 \times 10^0}$ | $\underline{0.936}$ | 3.454 | $\underline{0.924}$ |
| CF$^2$ | $7.82 \times 10^1$ | – | 0.019 | 3.757 | 0.015 |
| CLEAR | $2.15 \times 10^2$ | – | 0.037 | 27.35 | 0.037 |
| RSGG-CE | $\underline{3.49 \times 10^1}$ | $1.20 \times 10^2$ | 0.128 | $\underline{0.059}$ | 0.124 |
| GIST | $5.20 \times 10^1$ | $7.92 \times 10^0$ | **0.940** | 2.072 | **0.928** |

*Table 6.* Average test-set performance on BAShapes for 5-cross validations. The accuracy for the used GCN oracle in the test set is 99.9%.

|  | GED ↓ | Oracle Class ↓ | Validity ↑ | Sparsity ↓ | Fidelity ↑ |
|---|---|---|---|---|---|
| iRand | $\infty$ | $9.95 \times 10^1$ | 0.000 | $\infty$ | 0.000 |
| CF-GNNExp. | $9.09 \times 10^2$ | $\mathbf{0.20 \times 10^1}$ | 0.516 | 11.09 | $\underline{0.484}$ |
| CF$^2$ | $\infty$ | – | 0.000 | $\infty$ | 0.000 |
| CLEAR | $9.58 \times 10^2$ | – | $\underline{0.908}$ | 6.365 | 0.060 |
| RSGG-CE | $\mathbf{8.88 \times 10^1}$ | $1.03 \times 10^2$ | **1.000** | **0.575** | **0.968** |
| GIST | $\underline{1.11 \times 10^2}$ | $\underline{0.58 \times 10^1}$ | **1.000** | $\underline{0.816}$ | **0.968** |

*Table 7.* Average test-set performances on BBBP for 5-cross validations. The accuracy for the used GCN oracle in the test set is 92.2%.

|  | GED ↓ | Oracle Calls ↓ | Validity ↑ | Sparsity ↓ | Fidelity ↑ |
|---|---|---|---|---|---|
| iRand | $\mathbf{0.37 \times 10^1}$ | $\underline{1.42 \times 10^1}$ | 0.151 | **0.008** | 0.177 |
| CF-GNNExp. | $3.23 \times 10^2$ | $\mathbf{1.00 \times 10^0}$ | 0.931 | 5.227 | $\underline{0.784}$ |
| CF$^2$ | $1.34 \times 10^2$ | – | 0.208 | 3.350 | 0.178 |
| CLEAR | $1.43 \times 10^3$ | – | 0.267 | 11.27 | 0.164 |
| RSGG-CE | $\underline{2.33 \times 10^1}$ | $2.34 \times 10^2$ | 0.404 | $\underline{0.167}$ | 0.286 |
| GIST | $4.24 \times 10^1$ | $8.03 \times 10^1$ | **0.956** | 0.810 | **0.809** |

*Table 8.* Average test-set performances on BZR for 5-cross validations. The accuracy for the used GCN oracle in the test set is 96.1%.

|  | GED ↓ | Oracle Calls ↓ | Validity ↑ | Sparsity ↓ | Fidelity ↑ |
|---|---|---|---|---|---|
| iRand | $\underline{6.78 \times 10^1}$ | $2.93 \times 10^1$ | 0.332 | **0.029** | 0.146 |
| CF-GNNExp. | $6.23 \times 10^2$ | $\mathbf{0.10 \times 10^1}$ | **0.810** | 7.936 | **0.741** |
| CF$^2$ | $2.53 \times 10^3$ | – | 0.185 | $\underline{0.341}$ | 0.176 |
| CLEAR | $6.90 \times 10^1$ | – | 0.176 | 6.320 | 0.127 |
| RSGG-CE | $8.75 \times 10^1$ | $2.48 \times 10^2$ | $\underline{0.732}$ | 0.922 | $\underline{0.683}$ |
| GIST | $\mathbf{4.51 \times 10^1}$ | $\underline{0.74 \times 10^1}$ | **0.810** | 0.634 | **0.741** |

*Table 9.* Average test-set performances on COLORS-3 for 5-cross validations. The accuracy for the used GCN oracle in the test set is 27.7%.

|  | GED ↓ | Oracle Calls ↓ | Validity ↑ | Sparsity ↓ | Fidelity ↑ |
|---|---|---|---|---|---|
| iRand | $\mathbf{3.52 \times 10^1}$ | $1.03 \times 10^2$ | 0.392 | **0.043** | -0.007 |
| CF-GNNExp. | $4.35 \times 10^3$ | $\mathbf{0.10 \times 10^1}$ | 0.736 | 12.16 | 0.091 |
| CF$^2$ | $1.46 \times 10^2$ | – | 0.676 | 3.915 | 0.065 |
| CLEAR | $8.93 \times 10^3$ | – | 0.217 | 64.33 | 0.033 |
| RSGG-CE | $\underline{5.15 \times 10^1}$ | $3.70 \times 10^1$ | **0.884** | $\underline{0.291}$ | $\underline{0.147}$ |
| GIST | $1.41 \times 10^2$ | $\underline{0.43 \times 10^1}$ | **0.884** | 1.758 | **0.202** |

*Table 10.* Average test-set performances on ENZYMES for 5-cross validations. The accuracy for the used GCN oracle in the test set is 33.3%.

|  | GED ↓ | Oracle Calls ↓ | Validity ↑ | Sparsity ↓ | Fidelity ↑ |
|---|---|---|---|---|---|
| iRand | $\mathbf{0.79 \times 10^1}$ | $2.84 \times 10^1$ | 0.138 | **0.007** | 0.004 |
| CF-GNNExp. | $5.69 \times 10^2$ | $\mathbf{0.10 \times 10^1}$ | $\underline{0.910}$ | 4.947 | $\underline{0.077}$ |
| CF$^2$ | $9.38 \times 10^1$ | – | 0.437 | 1.524 | 0.017 |
| CLEAR | $9.34 \times 10^2$ | – | 0.370 | 26.23 | 0.040 |
| RSGG-CE | $\underline{4.96 \times 10^1}$ | $2.57 \times 10^2$ | 0.447 | $\underline{0.228}$ | 0.050 |
| GIST | $7.27 \times 10^1$ | $\underline{0.44 \times 10^1}$ | **0.970** | 0.957 | **0.203** |

*Table 11.* Average test-set performances on MSRC21 for 5-cross validations. The accuracy for the used GCN oracle in the test set is 91.2%.

|  | GED ↓ | Oracle Calls ↓ | Validity ↑ | Sparsity ↓ | Fidelity ↑ |
|---|---|---|---|---|---|
| iRand | $\mathbf{3.40 \times 10^1}$ | $1.88 \times 10^2$ | 0.035 | **0.004** | -0.035 |
| CF-GNNExp. | $2.98 \times 10^3$ | $\mathbf{0.20 \times 10^1}$ | **0.965** | 10.47 | 0.825 |
| CF$^2$ | $\underline{6.30 \times 10^1}$ | – | 0.018 | $\underline{0.230}$ | -0.018 |
| CLEAR | $1.23 \times 10^3$ | - | $\underline{0.933}$ | 4.296 | $\underline{0.855}$ |
| RSGG-CE | $1.03 \times 10^2$ | $4.78 \times 10^1$ | 0.912 | 0.329 | 0.807 |
| GIST | $2.20 \times 10^2$ | $\underline{0.52 \times 10^1}$ | **0.965** | 0.774 | **0.860** |

*Table 12.* Average test-set performances on PROTEINS for 5-cross validations. The accuracy for the used GCN oracle in the test set is 71.4%.

|  | GED ↓ | Oracle Calls ↓ | Validity ↑ | Sparsity ↓ | Fidelity ↑ |
|---|---|---|---|---|---|
| iRand | $\mathbf{0.20 \times 10^1}$ | $5.43 \times 10^2$ | 0.018 | **0.001** | -0.009 |
| CF-GNNExp. | $1.52 \times 10^3$ | $\mathbf{0.10 \times 10^1}$ | 0.378 | 6.273 | $\underline{0.202}$ |
| CF$^2$ | $5.94 \times 10^2$ | – | 0.039 | 11.00 | 0.018 |
| CLEAR | $6.02 \times 10^3$ | – | $\underline{0.563}$ | 703.8 | 0.051 |
| RSGG-CE | $\underline{4.39 \times 10^1}$ | $3.07 \times 10^2$ | 0.237 | $\underline{0.141}$ | 0.133 |
| GIST | $1.11 \times 10^2$ | $\underline{0.54 \times 10^1}$ | **0.791** | 1.463 | **0.425** |

## G.2. Full Panorama of SoTA Performances

We assess the performances of GIST and the SoTA methods according to all the metrics described in Appendix E. Table 5 – Table 12 illustrate these performances on the test-set for 5-cross validations on all the datasets. Notice how iRand and CF$^2$ on BAShapes (see Table 6) fail to produce valid counterfactuals (i.e., validity at 0); therefore, it is useless to measure the GED and sparsity of the produced candidates – hence, we report $\infty$.

## G.3. Spectra Combination $L(G^*) = \alpha L(G^\varepsilon) + (1 - \alpha)L(G)$

### G.3.1. SPECTRA COMBINATION WITH VARYING $\alpha$

The convex combination theorem for graph Laplacians provides a foundation for constructing intermediate graph structures by interpolating between the spectra of two graphs. Let $L(G)$ and $L(G^\varepsilon)$ denote the normalized Laplacians of graphs $G$ (e.g., a tree) and $G^\varepsilon$ (e.g., a cyclic graph), respectively. A convex combination of these Laplacians is defined in Equation (7). This equation guarantees that the spectrum of $L(G^*)$ lies within the convex hull formed by the spectra of $L(G)$ and $L(G^\varepsilon)$.

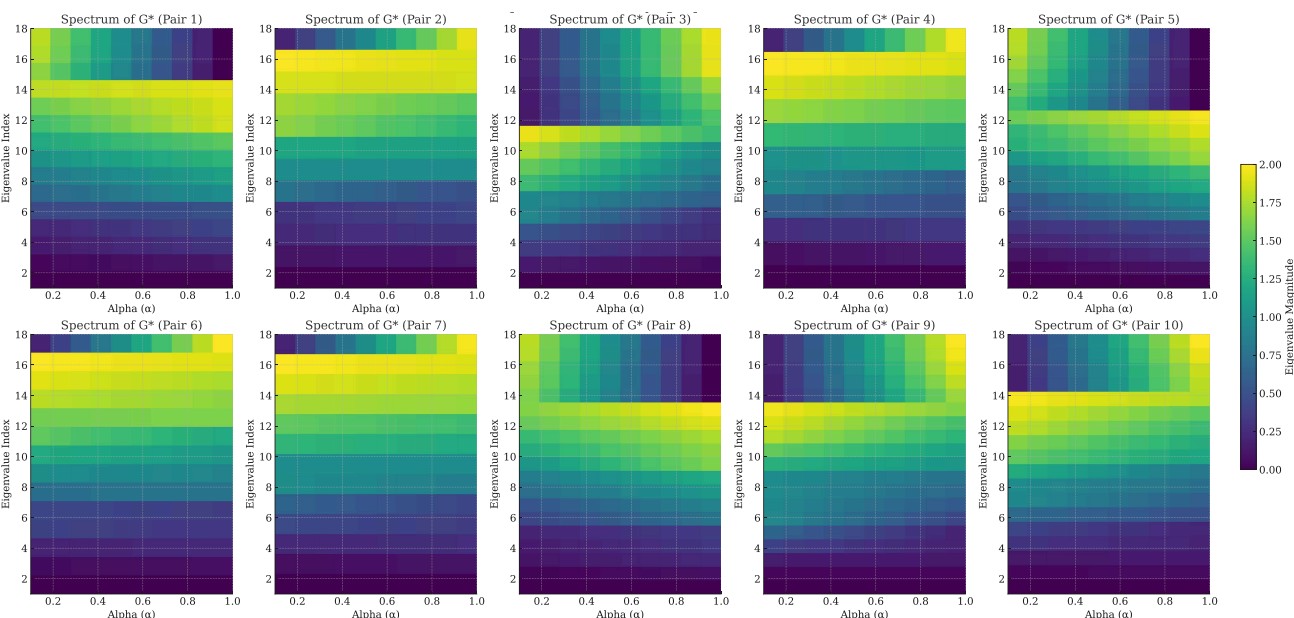

*Figure 9.* Heatmaps showing the spectra of $G^*$, obtained as a convex combination of the spectra of $G$ (tree) and $G^\varepsilon$ (cyclic graph) for 10 different pairs of $G$ and $G^\varepsilon$. The interpolation parameter $\alpha$ varies from 0.1 to 1, transitioning from the eigenvalues of $G$ ($\alpha = 0$) to those of $G^\varepsilon$ ($\alpha = 1$). Each plot demonstrates the smooth spectral transition governed by the convex combination.

As $\alpha$ varies, the spectrum of $G^*$ transitions smoothly between the spectral properties of $G$ ($\alpha = 0$) and $G^\varepsilon$ ($\alpha = 1$).

Figure 9 illustrates the spectra of $G^*$ for 10 different pairs of $G$ (trees) and $G^\varepsilon$ (cyclic graphs). Each heatmap corresponds to a unique pair, where the x-axis represents the interpolation parameter $\alpha$, ranging from 0.1 to 1. The y-axis indicates the eigenvalue index, while the color intensity reflects the magnitude of the eigenvalues. The leftmost region of each heatmap aligns with the spectrum of $G$ ($\alpha = 0$), and the rightmost region aligns with the spectrum of $G^\varepsilon$ ($\alpha = 1$). The gradual change in spectral intensities demonstrates the blending of structural properties as $\alpha$ increases. This figure underscores the potential of spectral interpolation to model continuous transitions between distinct graph structures, providing a tool for exploring intermediate configurations and understanding the impact of structural variations. Moreover, notice how with every $\alpha \in [0, 1]$, the connectivity property is maintained. In other words, because $G$ and $G^\varepsilon$ are connected graphs (either trees or cyclic graphs), then $G^*$ is also connected with $\lambda_1(L^{(G^*)}) = 0$ illustrated with dark blue.

### G.3.2. Spectral gap and connectedness of predicted $G^*$

Figure 10 shows the spectral gaps and the lowest eigenvalues for $G$, $G^\varepsilon$, and $G^*$ according to $\alpha = 0.9$ on AIDS (up left), COLORS-3 (up right), ENZYMES (down left), and PROTEIN (down right). We show log-scaled y-axes to highlight small differences in the prediction of $G^*$ via the loss Equation (12) – see triangles – and the optimal counterfactual obtained according to Definition 4.1 – see ×. Notice how, for each $(G, G^\varepsilon)$ pair, GIST finds $G^*$ whose spectral gap reports a negligible error of $\sim 1.75 \times 10^{-2}$ w.r.t. the optimal counterfactual. Moreover, GIST satisfies Theorem 4.5 since $\Delta(G)*$ is within the bounds of $\Delta(G)$ and $\Delta(G^\varepsilon)$. This intuition is also shown in terms of GED in Figure 3 and in terms of Frobenius norm differences in Figure 6 where $G^*$ is in-between $G$ and $G^\varepsilon$.

### G.3.3. Frobenius norm differences between $G$, $G^\varepsilon$ and $G^*$ according to Corollary 4.6

To support our claims that GIST produces spectrally similar counterfactuals, we illustrate (see Figure 11) the Frobenius norm differences presented in Corollary 4.6. Since GIST is learns the backtracking mechanism it might estimate $G^*$ that are not globally optimal, but rather locally. Therefore, we report a small estimation error of $5.62 \times 10^{-3}$ in terms of Frobenius norm difference where $G^*$ "prefers" to be closer to the structure of $G^\varepsilon$.

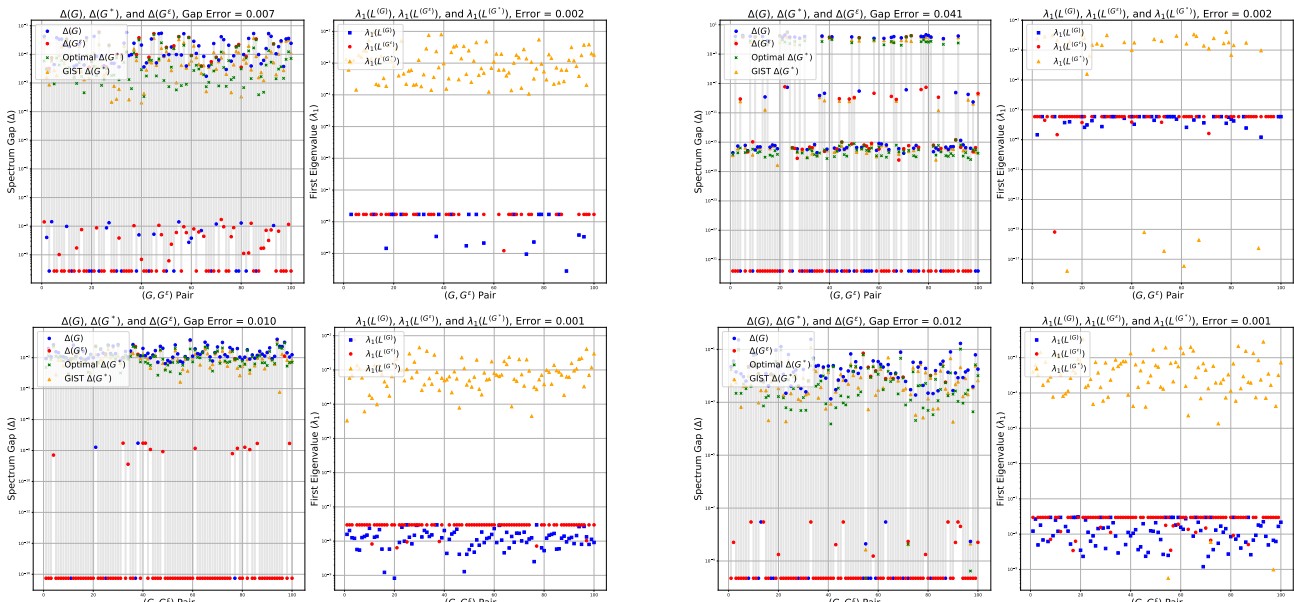

*Figure 10.* We show the spectral gaps of $G$, $G^\varepsilon$, and $G^*$ according to $\alpha = 0.9$ on 100 random samples of AIDS (up left), COLORS-3 (up right), ENZYMES (down left), and PROTEIN (down right). We also show the optimal counterfactual to assess the distance with the spectral gap of $G^*$. Each subplot on the right illustrates the first eigenvalue of $G$, $G^\varepsilon$, and $G^*$ to demonstrate the connectivity property in Theorem 4.4. To show zeros in log-scale, we correct them by adding the lowest gap/eigenvalue that is non-zero.

### G.4. Quantitative Analysis for $\alpha = 0.5$ vs $\alpha = 0.9$

To evaluate the sensitivity of GIST to the weighting parameter $\alpha$, we compared performance at $\alpha = 0.5$ (equal emphasis on content and style) against $\alpha = 0.9$ (content-dominant setting) across multiple benchmarks.

For the AIDS dataset, we computed the Frobenius norm between the expected and produced outputs as done in Figure 6 – see Figure 13. At $\alpha = 0.5$, the error reaches 0.537, which is over two orders of magnitude higher than the corresponding error at $\alpha = 0.9$. This substantial increase indicates that the model has difficulty reconciling competing objectives when neither is strongly prioritized.

Similarly, when emulating the setup from Figure 5 – see Figure 14 – the model yields an error of 0.013 at $\alpha = 0.5$, compared to only 0.005 at $\alpha = 0.9$. This further supports the observation that GIST performs more reliably when the optimization is skewed toward either content or style, rather than balanced equally.

These results demonstrate that GIST's performance degrades significantly under equal weighting conditions, and suggest that fine-tuning $\alpha$ toward task-specific priorities is crucial for optimal output quality.

### G.5. A Critique to SoTA papers using MUTAG and NCI1

We excluded all those datasets from TUDataset[9] that do not have any node attributes. As per GNNs message passing mechanism, the nodes share their feature vectors with their neighbors, hence then having meaningful embeddings. Given a graph $G = (X, A)$, GIST overshoots to $G^\varepsilon = (X^\varepsilon, A^\varepsilon)$ whose node features $X^\varepsilon$ go through TransConv layers. If $X^\varepsilon$ are missing, then the convolution layer does not produce anything meaningful to then estimate the edge probabilities (see Figure 2). To surpass this hurdle, we added 7 features regarding centralities: i.e., node degree, betweenness, closeness, harmonic centrality, clustering coefficient, Katz centrality, Laplacian centrality. In this way, at least we have something interesting to work with and not rely only on the topology of the graphs. Table 14 shows the performance against SoTA in terms of validity and fidelity on 5-fold cross-validations where the oracle is a 3-layer GCN with test accuracy of 86.8% for MUTAG.[10] Unfortunately, even after hyperparameter optimization was done on NCI1 with the introduced node features, any

---

[9]https://chrsmrrs.github.io/datasets/docs/datasets/

[10]We could not reproduce the 89.97% accuracy as in paperswithcode – https://paperswithcode.com/sota/graph-classification-on-mutag – with the parameters specified in the original paper of U2GNN.

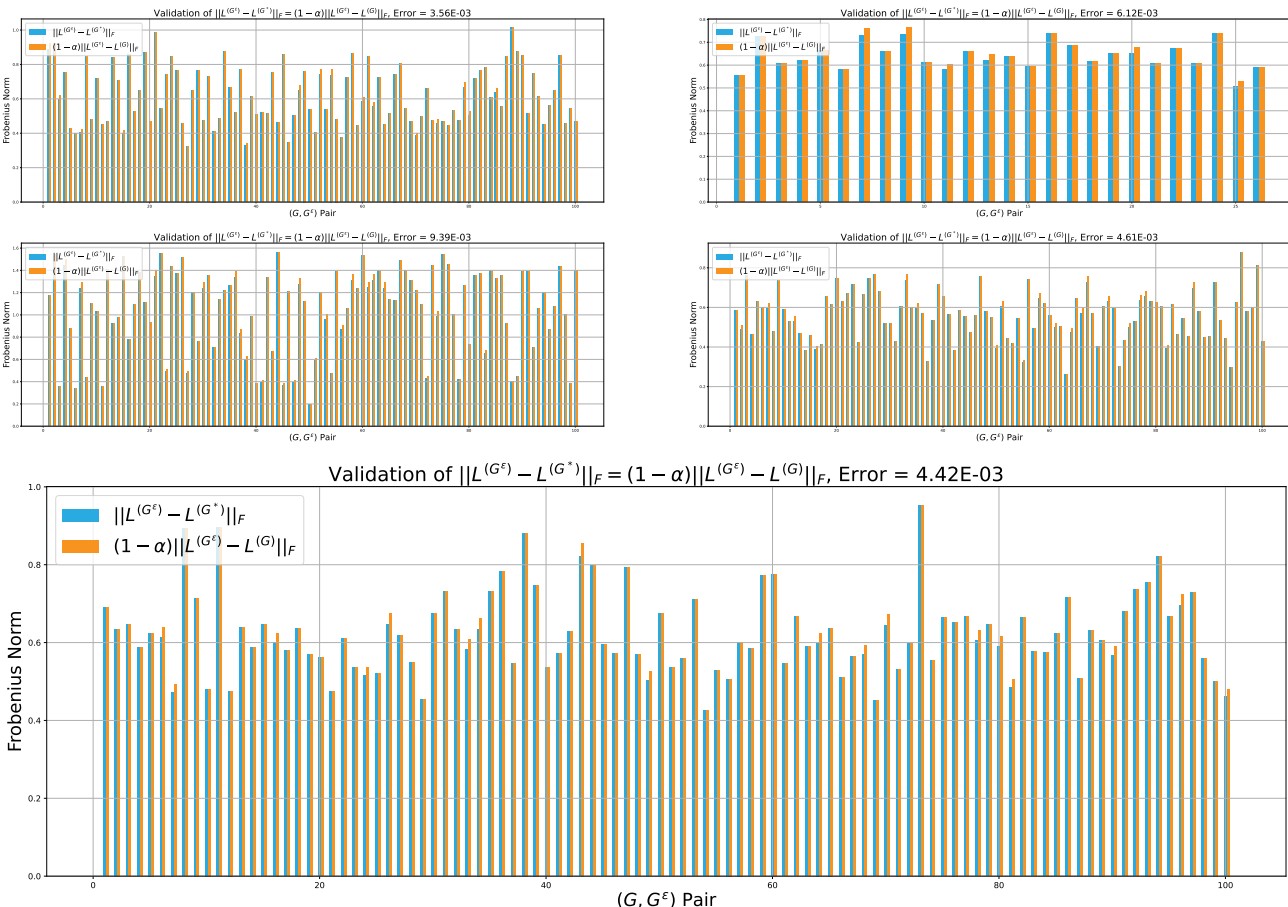

*Figure 11.* We show the Frobenius norm differences between the Laplacians of $G$, $G^\varepsilon$ and $G^*$ on AIDS (row=1, col=1), BZR (row=1, col=2), COLORS-3 (row=2, col=1), ENZYMES (row=2, col=2), and MSRC21 (row=3) for $\alpha = 0.9$.

kind of GCN (with any layer) and U2GNN (Nguyen et al., 2022) with the hyperparameter search space introduced in the original paper does not reach more than 40% of accuracy in the test set. We ran experiments with these oracles for NCI1, however the fidelity of the explainers was negative, which suggests that the explainers are actually doing adversarial attacks rather than explanations on the oracle (Prado-Romero et al., 2023). Hence, we decided to discard NCI1 and show only MUTAG. *We want to point out that these two datasets are not suitable for benchmarking purposes since, again, message passing mechanisms in GNNs rely on node feature aggregations on the neighbors. These two datasets do not have node features, and we are a puzzled how SoTA methods used them to compare against each other.*

### G.6. Runs on IMDB-M: GIST vs CLEAR

We report the results of 5-fold cross-validation on the IMDB-M dataset, as summarized in Table 13. Notably, we re-ran the CLEAR – the original paper that uses IMDB-M – from scratch and obtained a validity score of 0.45, which significantly deviates from the 0.96 originally reported, raising questions about the reproducibility and robustness of the original evaluation. Among all methods, GIST achieves the highest validity (0.87) and fidelity (0.17) w.r.t. 3-layer GCN oracle, which itself exhibits poor predictive performance with a test accuracy of only 48%.

The iRand baseline fails to generate any valid counterfactuals, rendering fidelity evaluation inapplicable. Furthermore, the uniformly low fidelity scores across all methods underscore a broader issue: the underlying GCN oracle performs poorly, limiting the interpretability and utility of generated counterfactuals. We also attempted to replicate the performance of U2GNN (Nguyen et al., 2022), a method claimed to be SoTA on IMDB-M, but achieved only 33% test accuracy using the same hyperparameters reported in the original paper – substantially lower than the 89.2% listed on paperswithcode. This discrepancy highlights the need for rigorous, transparent, and reproducible benchmarking practices in graph-based

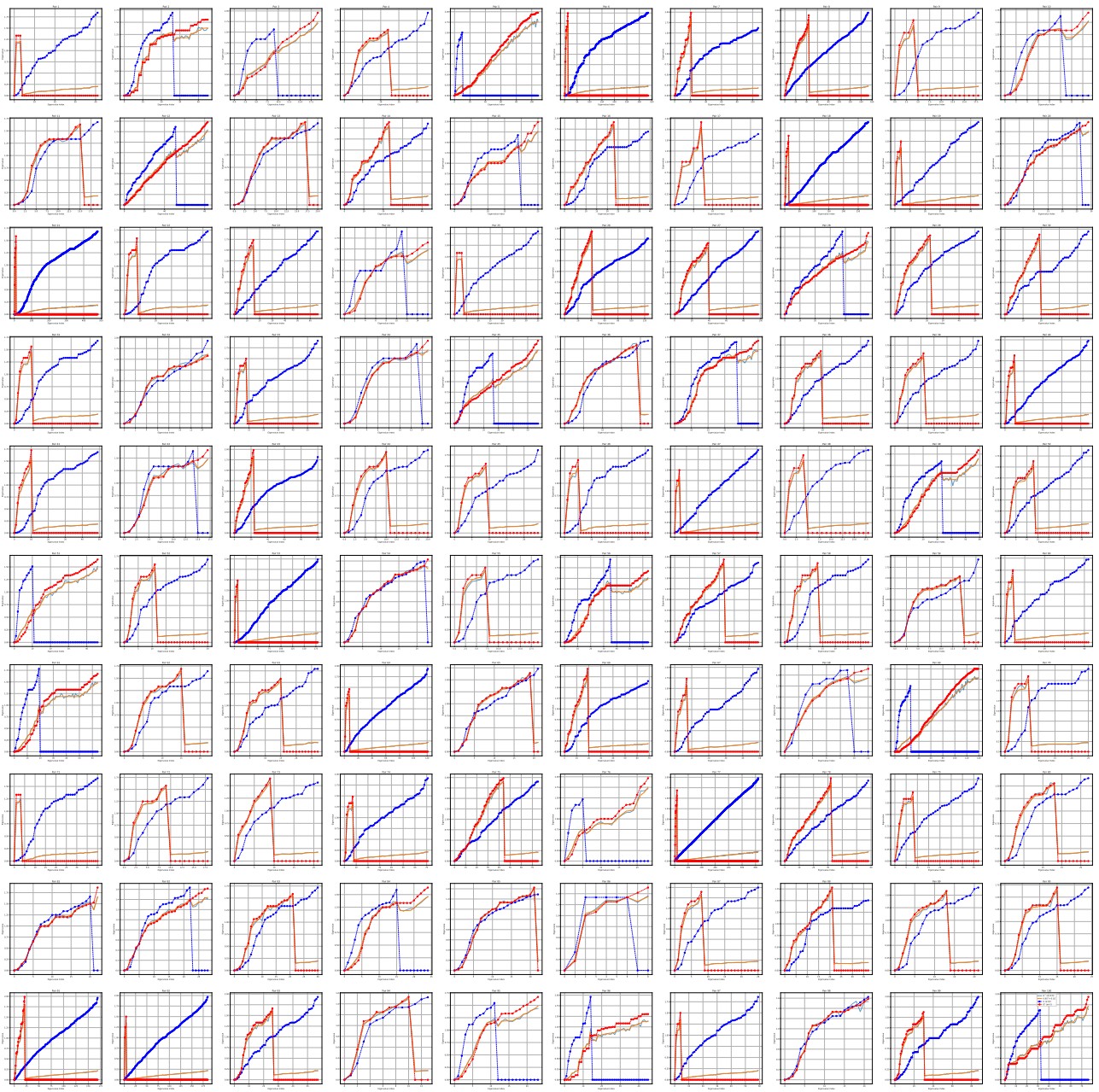

*Figure 12.* We show the eigenvalues of 100 randomly sampled instances on PROTEINS ($\alpha = 0.9$).

explainability research.

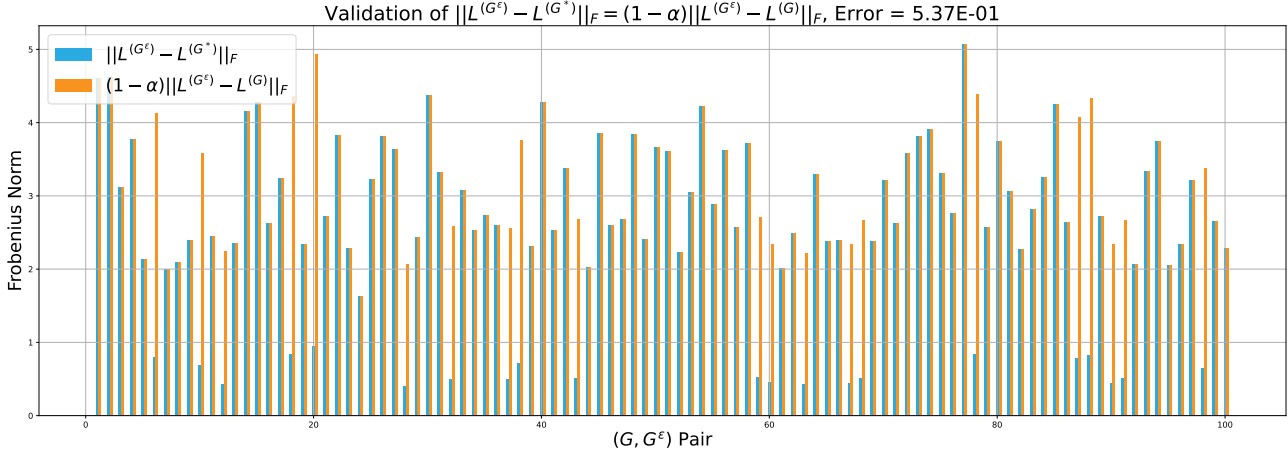

*Figure 13.* We show the Frobenius norm differences between the Laplacians of $G$, $G^\varepsilon$ and $G^*$ on AIDS for $\alpha = 0.5$.

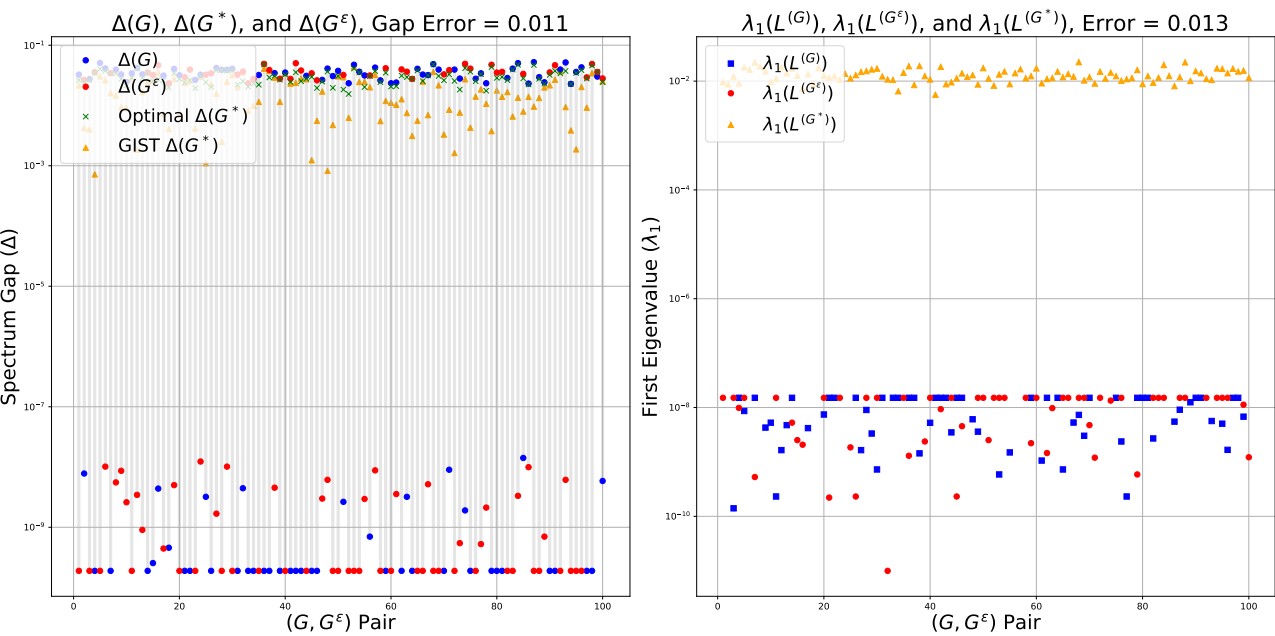

*Figure 14.* We show the spectral gaps of $G$, $G^\varepsilon$, and $G^*$ according to $\alpha = 0.5$ on 100 random samples of MSRC21. We also show the optimal counterfactual to assess the distance with the spectral gap of $G^*$. Each subplot on the right illustrates the first eigenvalue of $G$, $G^\varepsilon$, and $G^*$ to demonstrate the connectivity property in Theorem 4.4. To show zeros in log-scale, we correct them by adding the lowest gap/eigenvalue that is non-zero.

*Table 13.* Average test-set performance on IMDB-M for 5-cross validation. The accuracy of the GCN oracle in the test set is 48%.

|  | Validity | Fidelity |
| --- | --- | --- |
| iRand | 0.000 | – |
| CF-GNNExpl. | 0.670 | **0.170** |
| CF$^2$ | 0.710 | 0.090 |
| CLEAR | 0.450 | 0.050 |
| RSGG-CE | 0.690 | 0.090 |
| GIST | **0.870** | **0.170** |

*Table 14.* Average test-set performance on MUTAG for 5-cross validation. The accuracy of the GCN oracle in the test set is 86.6%.

|  | Validity | Fidelity |
| --- | --- | --- |
| iRand | 0.026 | 0.026 |
| CF-GNNExpl. | 0.447 | 0.237 |
| CF$^2$ | 0.026 | 0.026 |
| CLEAR | 0.921 | 0.395 |
| RSGG-CE | 0.947 | **0.737** |
| GIST | **1.000** | **0.737** |

