# OpenReview forum: "Graph Inverse Style Transfer for Counterfactual Explainability"
_ICML.cc/2025/Conference — ICML 2025 poster_

### Official Review · Reviewer_wac8 · 2025-03-05

**Overall Recommendation:** 3

**Summary:**

The authors introduce GIST their novel framework that generates counterfactual graph explanations. They leverage spectral style transfer to generate valid counterfactual explanations. Their architecture consists of two components: attention based node embeddings, and edge probabilities from the Gumbel-softmax trick.
After designing their algorithm they implement it on several real-world and synthetic datasets. They use the data sets BBBP, BZR, ENZYMES, MSRC21, PROTEINS, BA-SHAPES, and COLORS-3. They conduct experiments with respect to several SOTA baselines and compute relevant metrics such as validity and fidelity. They also conduct an ablation study on the interpolation factor which adjusts the distance from the decision boundary. They show a remarkable improvement in both validity and fidelity.

## update after rebuttal
Thanks for the authors' efforts in the rebuttal. I intend to keep my rating.

**Claims And Evidence:**

The claims in the paper are mostly correct. The claims are supported by theory and preliminary experiments that show the framework’s potentially ideal behavior.

**Essential References Not Discussed:**

paper [1] from Neurips 2021 is quite relevant and is neither referenced nor used as a baseline
[1] Bajaj, Mohit, et al. "Robust counterfactual explanations on graph neural networks." Advances in Neural Information Processing Systems 34 (2021): 5644-5655.

**Experimental Designs Or Analyses:**

Checked the design of the experimental setups and they are sound.

**Methods And Evaluation Criteria:**

The chosen datasets are good choices for their work. However, the authors should consider also adding additional datasets such as NCI1, MUTAG etc as these are common datasets. The authors also have left out a baseline method in counterfactual explanation methods [1].
[1] Bajaj, Mohit, et al. "Robust counterfactual explanations on graph neural networks." Advances in Neural Information Processing Systems 34 (2021): 5644-5655.

**Other Comments Or Suggestions:**

n/a

**Other Strengths And Weaknesses:**

The paper is well written, theoretically rigorous and well founded.

**Questions For Authors:**

See above issues

**Relation To Broader Scientific Literature:**

The paper does miss a reference in counterfactual graph explanations see [1]
[1] Bajaj, Mohit, et al. "Robust counterfactual explanations on graph neural networks." Advances in Neural Information Processing Systems 34 (2021): 5644-5655.

**Theoretical Claims:**

Briefly checked over theoretical claims and some proofs.

---

> ### Author Rebuttal · Authors · 2025-03-28
>
> We thank you for the effort made to review our paper, and for the nice score you chose to give it. Thank you for pointing out RCExplainer [2].
>
> **W1: Missed Bajaj reference**: We were aware of the paper, and decided to reviewed RCExplainer again to see whether we were missing something or not. We cross-validated it with what described in [1] specifically in Table 2 (page 12). As per [1], we confirm that it is a heuristic + learned-based method.
>
> In our related work section, we specifically stated that we concentrate on only learning-based and generative methods, hence the choice of the methods we compared against. However, we see the value of RCExplainer with its multiple learnt linear decision boundaries and then the search over these boundaries to find robust explainers, and will include it in our related work at camera ready.
>
> Unfortunately, the official code doesn't run (*the authors load some Huawei third party python packages that aren't available anywhere*) even after debugging and trying to port it to the GRETEL framework to have the same evaluation pipeline. Then, we also tried to run an unofficial code repository at https://github.com/idea-iitd/gnn-x-bench/blob/main/source/rcexplainer.py (we are not the authors so anonymity isn't breached if you want to take a look), however, the code doesn't support BBBP, BZR, ENZYMES, MSRC21, and COLORS-3 datasets. It's fine that it doesn't support COLORS-3 since this is multiclass and RCExplainer only does binary classification. However, not supporting the other listed datasets hinders us to compare it against other SoTA and GIST.
>
> *I believe a good compromise here is to recognize RCExplainer's validity as a heuristic+learned counterfactual explainer and list it in our related work section and highlight the fact that we only treat pure learned-based approaches*. What do you think?
>
> **Q1: Where are MUTAG and NCI1?**: We excluded all those datasets from TUDataset (https://chrsmrrs.github.io/datasets/docs/datasets/) that do not have any node attributes. As per GNNs message passing mechanism, the nodes share their feature vectors with their neighbors, hence then having meaningful embeddings. Given a graph $G=(X,A)$, GIST overshoots to $G^e=(X^e,A^e)$ whose node features $X^e$ go through TransConv layers. If $X^e$ are missing, then the conv. layer doesn't produce anything meaningful to then estimate the edge probabilities (see Fig. 2). To surpass this hurdle, we added 7 features regarding centralities: node degree, betweeness, closeness, harmonic centrality, clustering coefficient, Katz centrality, Laplacian centrality. In this way, at least we have something interesting to work with and not rely only on the topology of the graphs. Here are the performance against SoTA in terms of validity and fidelity on 5-fold cross-validations where the oracle $\Phi$ is a 3-layer GCN with test accuracy of 86.8% for MUTAG. Unfortunately, even after hypeparam optimization was done on NCI1 with the introduced node features, any kind of GCN (with any layer) and UGFormer [3] with the hyperparameter search space introduced in the original paper do not reach more than 40% of accuracy in the test set. We ran experiments with these oracles for NCI1, however the fidelity of the explainers was negative, which suggests that the explainers are actually doing adversarial attacks rather than explanations on the oracle [1]. Hence, we decided to discard NCI1 and show only MUTAG. *We want to point out that these two datasets aren't suitable for benchmarking purposes since, again, message passing mechanisms in GNNs rely on node feature aggregations on the neighbors. These two datasets don't have node features, and we are a bit puzzled how SoTA methods used them to compare against each other.*
> |  | Validity | Fidelity |
> |---|---|---|
> | CF$^2$ | 0.026$\pm$0.026 | 0.026$\pm$0.026 |
> | CF-GNNExp | 0.447$\pm$0.026 | 0.237$\pm$0.132 |
> | CLEAR | 0.921$\pm$0.026 | 0.395$\pm$0.079 |
> | iRand | 0.026$\pm$0.026 | 0.026$\pm$0.026 |
> | RSGG-CE | 0.947$\pm$0.000 | 0.737$\pm$0.158 |
> | GIST | **1.0$\pm$0.000** |**0.737$\pm$0.105**|
>
> [1] Prado-Romero et al. A Survey on Graph Counterfactual Explanations: Definitions, Methods, Evaluation, and Research Challenges. ACM CSUR 2024.
>
> [2] Bajaj et al. Robust counterfactual explanations on graph neural networks. NeurIPS'21
>
> [3] Nguyen et al. Universal graph transformer self-attention networks.WWW'22

---

### Official Review · Reviewer_xnpL · 2025-03-06

**Overall Recommendation:** 2

**Summary:**

GIST introduces a backtracking approach for graph counterfactual explainability using spectral style transfer. Unlike forward perturbation methods, it refines graphs to preserve global style and local content. GIST achieved excellent results experimentally.

**Claims And Evidence:**

Please refer to Strengths And Weaknesses.

**Essential References Not Discussed:**

Please refer to Strengths And Weaknesses.

**Experimental Designs Or Analyses:**

Please refer to Strengths And Weaknesses.

**Methods And Evaluation Criteria:**

Please refer to Strengths And Weaknesses.

**Other Comments Or Suggestions:**

None

**Other Strengths And Weaknesses:**

**Strengths**

- The paper presents a thorough spectral analysis, mathematically proving important properties including spectral gap bounds and Frobenius norm differences, which helps ensure the generated counterfactuals maintain coherence and semantic validity.

- GIST is evaluated across eight benchmark datasets, demonstrating its effectiveness in diverse graph settings.

**Weaknesses**

- The paper introduces an intermediary known graph G in the counterfactual generation process but does not explain how this graph is obtained or why it improves counterfactual accuracy. Since counterfactuals are based on assumptions without ground truth, the reliance on an intermediary known graph raises questions about its validity.

- The paper lacks a clear definition of what constitutes a counterfactual in different graph contexts. While structural transformations (e.g., spectral properties) are well-explained, the paper does not specify how counterfactuals are defined in user networks.

- The paper does not provide sufficient details on node embedding generation and how these embeddings change during counterfactual transformations.

- The paper evaluates on eight datasets but does not use commonly used benchmarks from prior counterfactual studies, such as Community and IMDB-M used in CLEAR [1]. It will be benefit to explain why the selected datasets are appropriate for evaluating graph counterfactuals.

[1] Ma, Jing, et al. "Clear: Generative counterfactual explanations on graphs." Advances in neural information processing systems 35 (2022): 25895-25907.

**Questions For Authors:**

Please check above.

**Relation To Broader Scientific Literature:**

Please refer to Strengths And Weaknesses.

**Theoretical Claims:**

Please refer to Strengths And Weaknesses.

---

> ### Author Rebuttal · Authors · 2025-03-25
>
> We thank you for the effort made to review GIST. It's unfortunate you didn't see its value during your original review. With the following, we tackle the weaknesses you mentioned, and hope to convince you of the paper's value.
>
> **W1: Intermediate graph $G^e$, and improved accuracy**: $G^e$ is obtained by overshooting the decision boundary of the oracle $\Phi$. This process is described in Sec. B and is referenced in Sec. 4. To reiterate, we take the dataset and divide the instances into sets containing of the same class according to $\Phi$. Then, for each input $G$ with label $y = \Phi(G)$, we take all the sets whose label isn't $y$, unify and shuffle them. Lastly, we pick the first instance from this shuffled set, and that's $G^e$. This simple yet effective mechanism guarantees the counterfactual begins from the correct region (i.e., $G^e$ is already on the other side of $\Phi$'s boundary w.r.t. the input $G$). *As a consequence, if we don't use the backtracking mechanism, and just return $G^e$ as the counterfactual, we'd have validity (aka explanation accuracy) equal to 1.* However, $G^e$ might be far from $G$, and we don't want that. That's why GIST walks back towards $G$ and minimizes the spectral difference (Fig 8 for Graph Edit Distance). By going back, GIST needs to learn not to recross $\Phi$'s boundary and get an invalid counterfactual produced, however this might happen due to poor boundary definition. We will acknowledge this limitation and make what mentioned above clearer at camera ready by allocating space via shrinking Sec. 5.3 as per **XATo**.
>
> **W3: Node embeddings**: I don't understand completely; I'll try my best here. Fig. 2 shows our architecture that takes in input a graph $G=(X \in \mathbb{R}^{n\times d},A)$ which is overshot to $G^e = (X^e \in \mathbb{R}^{n \times d},A^e)$. $X^e$ get then fed to the TransConv layers that project them to a latent space $\hat{X}^e \in \mathbb{R}^\ell$. The latent features are only used to estimate edge probabilities for the counterfactual candidate $G^*$. What I think you meant is whether the node embeddings in $\mathbb{R}^\ell$ are then injected on the nodes of the selected incident edges via the Gumbel + Bernoulli sampling. The short answer is "no". Thus, one can't trace how the embeddings are "transformed". However, a simple fix would do the trick: when we have $\hat{X}^e$, we can add a decoder network $h: \mathbb{R}^\ell → \mathbb{R}^d$ - e.g., learned GCN - to map the embeddings back to the input space, and train this network jointly with Eq. 12 by summing, for instance, $|X-g(\hat{X}^e)|_1$. In this way, we can see which embeddings contribute to the lowest difference (a desideratum of counterfactuality [1]) between $X$ and $g(\hat{X}^e)$. By doing this, we obtain a sparsity ↓ (Sec. E) as follows for org. GIST (up) and GIST with the embedding decoder (down). If you think this is valuable, we can add it at camera ready, in the appendix.
> |  | AIDS | BAShapes | BBBP | BZR | COLORS-3 | ENZYMES | MSCR21 | PROTEINS |
> |---|---|---|---|---|---|---|---|---|
> | $\|X - X^e\|_1$ | 2.07 | .82 | .81 | .63 | 1.76 | .96 | .77 | 1.46 |
> | $\|X - g(\hat{X}^e)\|_1$ | 1.36 | .63 | .50 | .41 | .97 | .74 | .38 | .83 |
>
> **W4: Community & IMDB-M**: Community was synthetically ad-hoc generated in CLEAR. This is why we can't reproduce the same dataset as in that paper. So, we opted to choose TreeCycles and BAShapes to cover RSGG-CE [2] and CF-GNNExp. [3] which were already supported in the GRETEL framework [4]. The results for IMDB-M on 5-fold cross-validations are here. Note that we ran CLEAR from scratch and the reported validity (0.45) isn't the one reported in the original paper (0.96). GIST is the best in terms of validity and fidelity w.r.t. a 3-layer GCN whose test accuracy is 48%. Here, we are using the version of GIST without node embeddings for consistency with what reported for the other datasets in the paper (Sec. G.2).
> |  | GIST | iRand | CF$^2$ | CLEAR | CF-GNNExp. | RSSG-CE |
> |---|:---:|:---:|:---:|:---:|:---:|:---:|
> |Validity|**0.87**|0| 0.71| 0.45| 0.67| 0.69|
> |Fidelity|**0.17**|-| 0.09| 0.05| 0.17| 0.09|
>
> iRand doesn't produce any valid counterfactuals, hence its fidelity cannot be measured. Also, all the methods have a low fidelity due to the oracle's "horrible" classification skills. We tried UGFormer [5], as the best SoTA in IMDB-M, however we got a test accuracy of 33% with the same hyperparams as in the original paper, instead of the reported 89.2% on paperswithcode.
>
> [1] Wachter et al. Counterfactual explanations without opening the black box: Automated decisions and the GDPR.
>
> [2] Prado-Romero et al. Robust stochastic graph generator for counterfactual explanations.AAAI'24
>
> [3] Lucic et al. Cf-gnnexplainer: Counterfactual explanations for graph neural networks.AISTATS'22
>
> [4] Prado-Romero & Stilo. Gretel: Graph counterfactual explanation evaluation framework.CIKM'22
>
> [5] Nguyen et al. Universal graph transformer self-attention networks.WWW'22

---

### Official Review · Reviewer_XATo · 2025-03-11

**Overall Recommendation:** 3

**Summary:**

The authors present a new method for generating counterfactual explanations for Graph Neural Networks (GNNs) based on an adaptation of neural style transfer. They then establish some theoretical results for the well-foundedness of their approach before presenting their method to learn the style transfer objective. Finally, the authors benchmark their method on synthetic and real-world datasets against current baselines.

**Claims And Evidence:**

The claims presented are supported by clear and convincing evidence.

**Essential References Not Discussed:**

The similarity metrics was not used to assess the method versus the baselines proposed, as decribed in [1].

[1] Counterfactual explanations and how to find them: literature review and benchmarking. Riccardo Guidotti, 2022.

**Experimental Designs Or Analyses:**

The experimental design is sound, and the analysis of part 5.2 and 5.3 is correct, although incomplete in my opinion, as a similarity metric is missing, which is key for counterfactual explanation (see essential references not discussed point).

**Methods And Evaluation Criteria:**

The method, datasets and baselines makes sense for the problem studied. The baselines and datasets are also relevant. However, I am not convinced by the choice of metrics, as the comparison with the baselines does not seem entirely fair.
You should include a measure of similarity, see [1].

[1] Counterfactual explanations and how to find them:
literature review and benchmarking. Riccardo Guidotti, 2022.

**Other Comments Or Suggestions:**

- In part 4.1, why do you assume that the Laplacian matrices commute? I know in the proof of theorem 4.4 you separate the cases commuting/non-commuting, but reusing the notation in part 4.1 is confusing.

- Put "BCE" in equation 6.

- Part 5: Clarify that you are backtracking from a dataset counterfactual.

- Part 5.3: in my opinion this part is too long. It echoes the basic interpolation results from 4.1, but it provides little in the characterization of counterfactual explanations.

- The proof of Weyl's inequality in the appendix is unnecessary.

- Overall, the proofs in Appendix A.1 should be much shorter (from 2 to 1/2 page), this is very basic linear algebra.

- Appendix A.2 is also very slow for an obvious property, only step 5 is necessary.

- Appendix A.3 : Another direct simple application of Weyl's inequality, this is also very slow. Note that step 1 is poorly formulated.

- Appendix A.4 should be a one liner.

- Overall, the theoretical insights claim appears weak. You are just interpolating between two graphs. The analysis proves that, indeed, your framework interpolates between two graphs in terms of eigenvalues and norm. How does it relate to counterfactual explanation?

- Appendix 2: I don't understand equation (50). What is the minimum of a shuffled set? Also $Q$ is not define

**Other Strengths And Weaknesses:**

Strengths:

- The idea is interesting.
- The writing is good.
- The experiments are extensive and thorough.

Weaknesses:
- In my opinion, the modest theoretical insights do not really add any interpretability into the counterfactual explanations. It does, however, show that you can interpolate between two graphs.

- A similarity metric should be used to compare the baselines, and ideally would be addressed in the objective.

- Some weaknesses and limitations of the proposed method should be discussed in the paper.

**Questions For Authors:**

- definition 4.1: I am puzzled about your definition of style and content. How do you motivate using the Laplacian of graphs to define style?

- definition 4.1: Why not take the style between $G^*$ and $G^\varepsilon$ and the content between $G^*$ and $G$?

- Part 5.1 and Appendix C: for the parameters of the other explainers, wouldn't a fair comparison try to optimize their parameters for validity?

- I am skeptical about your choice of metrics; would not $\alpha = 1$ achieve perfect validity and very good fidelity, as you would simply return the dataset counterfactual? Then what is the point of the analysis in 5.2?

- Part 5, Figure 4: Why are there so many zeros in those graphs? Is this related to the difference in the sizes of $G$ and $G^\varepsilon$?

- Part 5: Can you give some spectral analysis for $\alpha = 0.5$? This choice of parameter may make it harder for your method to perform according to the theoretical results.

- Part 5: How is your method fairing compared to the baselines wrt a similarity metric (eg. GED for example)?

- I think an interesting direction could be to look into incorporating the GNN prediction to adjust the alpha parameter, hence actually balancing the similarity and validity metric in your objective. Have you considered such improvements?

**Relation To Broader Scientific Literature:**

The work seems interesting, although it is difficult to compare their method with the chosen baselines, since its focuses primarily on the Validity metric.

**Theoretical Claims:**

I have checked the correctness of all the proofs of the lemmas and theorems in the paper.

---

> ### Author Rebuttal · Authors · 2025-03-30
>
> Oh, this is awesome. Thanks for reading the paper and not using an LLM to generate your review :-)
>
> Your suggestions will better our paper. The proofs will be much shorter and clarifications on counterfactuality more thorough. Sec. 5.3 contains the two most important evaluation metrics as per [1]. We'll put the GED (now in Sec.G.1) in the main paper.
>
> **Just interpolation; no insights for counterfactuals**: While the literature advocates for "hard-core" generative models, GIST learns to backtrack from a dataset example $G^e$ toward $G$. We agree that the main contribution is interpolation, but, it's applicability is suitable for counterfactuality as shown in the experiments. Interpolating $G$ and $G^e$ guarantees $G^*$ is in-distribution, which promotes plausability [2]. We'll clarify this better at camera ready, and measure plausability.
>
> **Why Laplacian?**: Laplacians capture global structural patterns, e.g., connectivity and symmetry, that are largely invariant to specific node identities. This follows a similar rationale to neural style transfer in images, where Gram matrices of feature activations are used to model style since they encode correlation patterns among features rather than spatial arrangements. Additionally, using the Laplacian aligns with prior work in spectral graph theory, where the eigenvalues and eigenvectors of the Laplacian are shown to be robust descriptors of global structure, and have been used in graph matching [3] and generation [4].
>
> **Eq. 50?**: $Q: \mathcal{G}\times \Phi→\mathcal{G}$ (we missed this). We put $\arg \min$ to emulate a for-loop on $U$. This means we take the first $G^e \in U$. In hindsight this is also confusing to us. We'll change it to $k^*=\min [ k\in[1,n]|\Phi(G)\neq \Phi(G_k) ]\; \forall G_k \in U$, and $G^e = G_{k^*}$
>
> **Similarity metric, objective func and GED**: By using $G$ as a pulling factor governed by $\alpha$ to produce $G^*$, the similarity is already addressed in the objective although not explicitly as in the desideratum in [2]. In principle, the more $G^*$ goes toward $G$, the lower the edit distance should be, as well as the validity (Fig. 3). We do report GED (Fig. 8, Tab 5-12): GIST is better than CF-GNNExp (our main competitor in validity), though it's not the best across the board due to our choice of $\alpha = 0.9$.  Lower $\alpha$ would improve similarity at the cost of validity, a trade-off we make explicit.
>
> **SoTA hyperparams**: We used the hyperparams reported in their original papers. We can optimize them for validity, but this would require months of optimizations. We can optimize them for one dataset (e.g., BAShapes) and use the same for the rest. Do you think this is a good compromise?
>
> **Sec. 5.2 and $\alpha=1$**: We aim to balance validity and similarity. Setting $\alpha=1$ would ignore the counterfactual signal, leading to perfect validity but poor GED, as $\mathcal{L}_{style}$​ becomes zero, and no learning (backtracking) is happening. Our formulation indirectly encourages similarity through $\alpha$, which controls how much $G$ influences $G^*$. The analysis in Sec. 5.2 is necessary to demonstrate how this trade-off affects GED, even if it's not an explicit loss term.
>
> **Fig. 4**: The zeros are indeed paddings to compute the spectral differences, which need the adj and degree matrices to be of same dimensions. To account for different graph sizes, we could use the Wasserstein distance of the eigenvalues instead of L1. This would need further theoretical investigation. We'll clarify this in the figure's caption.
>
> **Spectral analysis for $\alpha=0.5$**: Your intuition is right; when $\alpha=0.5$, GIST struggles between preserving content and matching style. We computed the frobenius norms (expected vs. produced) as in Fig. 6 for $\alpha=0.5$ on AIDS and get an error of 0.537, two orders higher than with $\alpha=0.9$. We also emulate Fig. 5 with $\alpha=0.5$ and obtain an error of 0.013 instead of 0.005 as with $\alpha=0.9$. Lastly, for Fig. 4 we have an error of 0.043 instead of $2.051\times10^{-3}$. Unfortunately, we can't show images here, but we'll include this analysis in the appendix, and mention it in a new limitation section.
>
> **Switch content and style**: Taking style from $G^e$  and content from $G$ would attempt to generate a graph structurally similar to a perturbed counterfactual but semantically identical to the original, which defeats the purpose of generating counterfactuals, since semanticity usually drives $\Phi$'s prediction. GIST is designed to minimally deviate from $G$ structurally, but still flip the class as in $G^e$.
>
> [1] Prado-Romero et al. A survey on graph counterfactual explanations: definitions, methods, evaluation, and research challenges.CSUR'23
>
> [2] Guidotti.Counterfactual explanations and how to find them: literature review and benchmarking.
>
> [3] Yan et al. A short survey of recent advances in graph matching.ICMR'16
>
> [4] Dwivedi et al. Benchmarking Graph Neural Networks.JMLR'23

---

### Official Review · Reviewer_6kC8 · 2025-03-14

**Overall Recommendation:** 3

**Summary:**

This paper introduces Graph Inverse Style Transfer (GIST), a novel framework for counterfactual explainability in graph neural networks (GNNs). Unlike traditional forward perturbation-based counterfactual methods, GIST employs a backtracking mechanism inspired by style transfer in vision. By first overshooting a decision boundary and then refining the counterfactual graph to align with the original graph’s spectral properties, GIST aims to generate semantically valid and structurally faithful counterfactuals. The method is evaluated on eight benchmark datasets, where it demonstrates an increase in counterfactual validity and improvement in fidelity compared to state-of-the-art approaches.

## update after rebuttal
Thank the authors for response. Some of my concerns have been addressed, I intent to maintain my originial score.

**Claims And Evidence:**

Most claims in the submission are well-supported.

**Essential References Not Discussed:**

N/A

**Experimental Designs Or Analyses:**

The experiments could be improved with more classical and SOTA baselines, and include more complex real-world graphs for in-depth evaluation. Adding further user case studies for qualitative assessment would also be beneficial.

**Methods And Evaluation Criteria:**

Yes, the method and evaluation are in a decent design and suitable for the studied problem.
But it would be beneficial to include more baselines (as this field has been well studied in the past a few years).

**Other Comments Or Suggestions:**

N/A

**Other Strengths And Weaknesses:**

The paper introduces a novel and well-theorized counterfactual framework for GNNs. But there are some other potential concerns here to address:

- While the experimental results show improvements over baselines, a more detailed analysis of why GIST outperforms existing methods—beyond just numerical results—would make the contributions clearer.
- More specific user studies would be beneficial to include.

**Questions For Authors:**

- How scalable is GIST in large-scale real-world graphs?
- How does GIST compare to causal-based counterfactual approaches rather than just perturbation-based baselines?
- Have you considered potential applications in real-world case studies?

**Relation To Broader Scientific Literature:**

The proposed work relates to general counterfactual explainability in GNNs.

**Theoretical Claims:**

The provided theory makes sense at a high level.

---

> ### Author Rebuttal · Authors · 2025-03-25
>
> We thank you for the effort made to review our paper, and for the nice score you chose to give it. With the following, we hope to answer your questions, and convince you of the value of GIST.
>
> **W2: Specific user studies.** We want to point out that the scope of this paper was not to involve users (in fact, none of them are present in the paper) in assessing the goodness of the produced counterfactuals since graph counterfactuals can be quite complicated to illustrate, especially when many edges/nodes are being added/removed from the original instance (see **W1** for a proposal of qualitative analysis).
>
> **W1: Detailed analysis of GIST vs. SoTA beyond numerical results.** As per [3] - the most-up-to-date survey in graph counterfactuality - all methods, besides those designed specifically for molecule explanations where illustrations are possible and very clean visually, use the quantitative metrics we used (see Sec. E & G.2). However, we found in the literature (e.g., [1,2] seem to be consistent in styling) an interesting and visually-pleasing way to show the differences between the original instance and counterfactual produced by illustrating their adjacency matrices. We will extend this way of visualizing also for node feature perturbations since GIST supports this mechanism, differently from the abovecited papers. We believe this would add value to the quantitative measures we provided in the paper, and would appreciate your input in this regard.
>
> **Q1: How scalable is GIST in large-scale real-world graphs?**
>
> In Sec. F, we have a time complexity analysis that treats both densely- and sparsely-memorized graphs. To reiterate, since GIST needs to find an eigenvalue decomposition to compute the Laplacian, for dense graphs we have a complexity of $\mathcal{O}(n^3)$ where $n$ is the number of nodes in a graph, and for sparse graphs it amounts to $\mathcal{O}(k(n+m))$ where $m$ is the number of edges and $k$ is the number of eigenvalues to find. Since $k$ is a constant, then this amounts to $\mathcal{O}(n+m)$.  Since our implementation uses sparsely-memorized graphs, and from Table 3 in Sec. D you can see that most graphs are sparse anyway ($m<<n^2$ on avg), the execution time is linear on the number of nodes $n$ and edges $m$. Thus, scaling GIST to huge graphs is not expensive, even in real-world graphs that are generally sparse.
>
> **Q2: GIST vs. causal-based explainers:**  To the best of our knowledge, only CLEAR [4] is a causal-based explainer. We included it on our experiments and GIST consistently performs better across the board in validity (Tab. 1) and fidelity (Tab. 2) and counterfactual similarity with the input in terms of Graph Edit Distance (see Fig. 8). This is also because CLEAR, when producing a counterfactual candidate, it gives a fully-connected stochastic graph which then it needs to match to the input. Note, that graph matching is an NP-hard problem and there are some approximations which undermine CLEAR's performances. GIST, on the other hand, is very simple and understandable: *overshoot the oracle $\Phi$'s decision boundary and backtrack via graph transformers*.
>
> **Q3: Real-world applications.** One compelling real-world application of GIST is drug repurposing. Suppose we have a known drug A that effectively treats disease $d_1$. Our goal is to discover a new drug B that treats a different disease $d_2$, potentially by modifying A. However, directly identifying B is often difficult and financially expensive. GIST could help by first exploring compounds that are far from A - those that may not initially preserve A’s chemical properties but show efficacy against $d_2$ (albeit with more side effects). From there, GIST can iteratively "walk back" toward A, gradually optimizing for lower side effects while preserving therapeutic relevance to $d_2$. This path - from distant molecular candidates back toward A - could lead to a novel compound B that is both effective for $d_2$ and safer. We find this direction highly promising and plan to investigate it further. However, applying GIST to drug discovery requires close collaboration with chemists and pharmaceutical experts, as well as human-in-the-loop evaluation to ensure the physical plausibility of generated compounds. Such interdisciplinary work would also naturally address the need for domain-specific user studies, which is a point you raised in your original review.
>
> [1] Prado-Romero et al. Robust stochastic graph generator for counterfactual explanations. AAAI'24 (page 15-16, Figs. 10-11 of the suppl. material on arXiv)
>
> [2] Prenkaj et al. Unifying Evolution, Explanation, and Discernment: A Generative Approach for Dynamic Graph Counterfactuals. KDD'24 (Figs. 7-8)
>
> [3] Prado-Romero et al. A survey on graph counterfactual explanations: definitions, methods, evaluation, and research challenges. ACM CSUR'24.
>
> [4] Ma, et al. CLEAR: Generative counterfactual explanations on graphs. NeurIPS'22.

---

### Decision · Program_Chairs · 2025-05-01

**Decision:**

Accept (poster)

**Comment:**

The reviewers generally acknowledged that the proposed approach was novel and reasonable.

However, the reviewers still felt that the paper lacked clarity on how and why the process (particularly selecting $G^\epsilon$ and interpolating) is good for generating counterfactuals. This seems to be in part because it is unclear what a good counterfactual would be in different contexts. For example, what does a meaningful counterfactual mean in user networks?  Is it based on outcome label changes (e.g., loan approval vs. rejection), demographic information, or other characteristics? This lack of clarity makes it difficult to evaluate the practical relevance of the method.

Also, the paper does not seem to properly address the interaction between node features and graph structure during counterfactual generation. Real-world counterfactuals often involve complex interactions where changes in node attributes (e.g., changing a student's gender) would naturally affect their connections (e.g., friendship based on different interests). This interdependence between node features and graph structure is a critical consideration in counterfactual generation that appears overlooked [1,2].

Given this, I recommend that the authors revise their paper to more clearly explain what makes a good graph counterfactual in different contexts and then connect that definition with their proposed method. Why will this process produce better counterfactuals w.r.t. the specifically desired qualities of a graph counterfactual?

[1] Ma, Jing, et al. "Learning fair node representations with graph counterfactual fairness." Proceedings of the fifteenth ACM international conference on web search and data mining. 2022.

[2] Ling, Hongyi, et al. "Counterfactual Fairness on Graphs: Augmentations, Hidden Confounders, and Identifiability." (2024).